# Neural network and kinetic modelling of human genome replication reveal replication origin locations and strengths

**Jean-Michel Arbona** [1]*, **Hadi Kabalane** [2]¤, **Jeremy Barbier** [2], **Arach Goldar** [3], **Olivier Hyrien** [4]*, **Benjamin Audit** [2]*

1 Laboratoire de Biologie et Modélisation de la Cellule, ENS de Lyon, Lyon, France, 2 ENS de Lyon, CNRS, Laboratoire de Physique, Lyon, France, 3 Université Paris-Saclay, CEA, CNRS, Institute for Integrative Biology of the Cell (I2BC), Gif-sur-Yvette, France, 4 Institut de Biologie de l'Ecole Normale Supérieure (IBENS), Ecole Normale Supérieure, CNRS, INSERM, Université PSL, Paris, France

¤ Current address: Soladis Group, Lyon, France
* jeanmichel.arbona@ens-lyon.fr (J-MA); olivier.hyrien@bio.ens.psl.eu (OH); benjamin.audit@ens-lyon.fr (BA)

## Abstract

In human and other metazoans, the determinants of replication origin location and strength are still elusive. Origins are licensed in G1 phase and fired in S phase of the cell cycle, respectively. It is debated which of these two temporally separate steps determines origin efficiency. Experiments can independently profile mean replication timing (MRT) and replication fork directionality (RFD) genome-wide. Such profiles contain information on multiple origins' properties and on fork speed. Due to possible origin inactivation by passive replication, however, observed and intrinsic origin efficiencies can markedly differ. Thus, there is a need for methods to infer intrinsic from observed origin efficiency, which is context-dependent. Here, we show that MRT and RFD data are highly consistent with each other but contain information at different spatial scales. Using neural networks, we infer an origin licensing landscape that, when inserted in an appropriate simulation framework, jointly predicts MRT and RFD data with unprecedented precision and underlies the importance of dispersive origin firing. We furthermore uncover an analytical formula that predicts intrinsic from observed origin efficiency combined with MRT data. Comparison of inferred intrinsic origin efficiencies with experimental profiles of licensed origins (ORC, MCM) and actual initiation events (Bubble-seq, SNS-seq, OK-seq, ORM) show that intrinsic origin efficiency is not solely determined by licensing efficiency. Thus, human replication origin efficiency is set at both the origin licensing and firing steps.

## Author summary

DNA replication is a vital process that produces two identical replicas of DNA from one DNA molecule, ensuring the faithful transmission of genetic information from mother to daughter cells. The synthesis of new DNA strands initiates at multiple sites, termed replication origins, propagates bidirectionally, and terminates by merging of converging

**Data Availability Statement:** All the data and code necessary to reproduce this work are availabel at https://github.com/organic-chemistry/repli1D.

**Funding:** This work was supported by the Agence Nationale de la Recherche (ANR-18-CE45-0002 and ANR-19-CE12-0028) and the Cancéropôle Ile-de-France and the INCa (PL-BIO16-302). OH was also supported by the Ligue Nationale Contre le Cancer (Comité de Paris; RS19/75-75), the Association pour la Recherche sur le Cancer (PJA 20171206387) and the Fondation pour la Recherche Médicale (FRM EQU202203014910). The funders had no role in study design, data collection and analysis, decision to publish, or preparation of the manuscript. Hadi Kabalane and Jeremy Barbier received a salary from the ANR.

**Competing interests:** The authors have declared that no competing interests exist.

strands. Replication initiation continues in unreplicated DNA but is blocked in replicated DNA. Experiments have only given partial information about origin usage. In this work we reveal the exact propensity of any site to initiate replication along human chromosomes. First, we simulate the DNA replication process using approximate origin information, predict the direction and time of replication at each point of the genome, and train a neural network to precisely recover from the predictions the starting origin information. Second, we apply this network to real replication time and direction data, extracting the replication initiation propensity landscape that exactly predicts them. We compare this landscape to independent origin usage data, benchmarking them, and to landscapes of protein factors that mark potential origins. We find that the local abundance of such factors is insufficient to predict replication initiation and we infer to which extent other chromosomal cues locally influence potential origin usage.

## 1 Introduction

In eukaryotes, chromosome replication starts at multiple sites referred to as replication origins [1]. Origins are licensed for replication during the G1 phase of the cell cycle, when the origin recognition complex (ORC) loads the MCM2–7 replicative helicase in an inactive, double hexameric ring form (MCM DH), around origin DNA [2–5]. This symmetric configuration prepares the helicases to initiate bidirectional replication upon activation. Origin activation (or firing) can take place at different times through S phase, by binding of multiple firing factors that trigger origin DNA unwinding, convert the inactive MCM DH into two active Cdc45/MCM/GINS helicases that each encircles and translocates 3'-to-5' along a single DNA strand, and recruit DNA polymerases and accessory factors for processive replication [6]. Only a fraction of MCM DHs lead to productive initiation events, while the rest is inactivated by passing replication forks originating from other origins. This origin passivation mechanism [7] cooperates with MCM2–7 loading restriction to G1 phase to prevent rereplication in a single cell cycle [8].

Several experimental techniques allow to monitor origin licensing and firing as well as replication progression during S phase. Origin licensing can be monitored by experimental detection of ORC and MCM proteins, whose profiles are highly though not perfectly concordant [9–12]. In contrast to these potential origin profiles, actual initiation events can be monitored by sequencing purified replication initiation intermediates, such as short nascent DNA strands (SNS-Seq; [13]) or bubble-containing restriction fragments (Bubble-Seq; [14]). These two methods are only weakly concordant [15, 16]. Other methods monitor replication progression along the genome. Mean replication timing (MRT) profiles have been computed by sequencing newly replicated DNA from sorted cells at different stages of S phase (Repli-seq; [17–19]) or by determining DNA copy number from proliferating cells [20]. Peaks of early MRT must contain origins, but low resolution (50–100 kb; [17, 18]) has long precluded precise origin mapping from human MRT profiles. Replication fork directionality (RFD) profiles, obtained by strand-oriented sequencing of purified Okazaki fragments (OK-seq) were more resolutive (< 5 kb) [21, 22]. Okazaki fragments mapping to the Watson and Crick strands are generated by leftward-(L) and rightward-(R) moving forks, respectively. OK-seq therefore reveals the proportions of R and L forks at any locus in a cell population. RFD is defined as the difference between the local proportions of R and L forks. RFD varies from -1 to 1, with these extreme values indicating that 100% of the forks are moving leftward or rightward, respectively. RFD profiles revealed that: (i) each cell line contains 5,000—10,000 broad (10–100 kb) initiation

zones (IZs), characterised by a left-to-right upward shift in RFD; (ii) IZs often but not always flank active genes; (iii) termination events occur in broad zones (TZs), characterised by a downward RFD shift; (iv) TZs can be directly adjacent to IZs or separated from them by extended regions of unidirectional replication; (v) large randomly replicating regions, characterised by extended segments of null RFD, are observed in silent heterochromatin. OK-seq IZs were confirmed genome-wide by EdUseq-HU [23], high-resolution Repli-Seq [19], Optical Replication Mapping (ORM) [24] and by replicative DNA polymerase usage mapping (Pu-seq) [25]). Importantly, initiation events may additionally occur outside IZs, but in a too dispersed manner to be directly detected in cell population profiles. Recent single-molecule and OK-seq analyses of the yeast genome [26, 27], of two model chicken loci [28], and ORM analysis of the human genome [24] provided direct evidence for dispersed initiation between efficient IZs in these systems.

IZs can be shared between cell types or specific to a cell type, suggesting epigenetic regulation. They are enriched in DNAse I hypersensitive sites (HSSs) and histone modifications or variants such as H3K4me1, H3K27ac and H2A.Z, that usually mark active transcriptional regulatory elements [21, 22, 29]. H2A.Z was proposed to facilitate origin licensing and firing by recruiting SUV420H1, which promotes H4K20me2 deposition, in turn facilitating ORC binding [30]. Furthermore, binding sites for the firing factor MTBP were found to colocalize with H3K4me1, H3K27ac, H2A.Z, and other active chromatin marks [31].

What mechanisms could regulate origin firing? Modelling studies showed that a probabilistic interaction of potential origins with rate-limiting firing factors, engaged with forks and recycled at termination events, can predict the time-dependent profiles of origin firing rate and fork density universally observed in eukaryotes [7, 32, 33]. Experimental studies indeed suggested that rate-limiting activators regulate replication kinetics in yeast [34, 35] and metazoans [36, 37]. Thus, a simple model for replication regulation is that potential origins fire at different mean times because of their different affinities for limiting factors [38]. Alternatively, potential origins may all have the same affinity for firing factors but their variable density along the genome may determine MRT [11, 39, 40]. We refer to these two distinct models as the origin affinity model and the origin density model, respectively.

Modelling studies indicate that the reproducible spatial structure of genomic replication profiles can emerge from stochastic firing of individual origins [41, 42]. Gindin et al [42] built a kinetic model in which the time-dependent probability of initiation at a yet unreplicated site was the product of the time-dependent availability of a limiting factor by the time-independent, local value of a genomic "initiation probability landscape" (IPLS). Of the various genomic and epigenomic profiles used as estimates of IPLS, DNase I HSS profiles produced the best match with experimental MRT profiles (Pearson correlation between simulated and experimental MRT of 0.865). Importantly, the same IPLSs did not produce realistic MRT profiles in models that did not include competition for limiting fork-associating factors [42]. Since this model did not explicitly separate origin licensing and firing, however, it remained unclear whether the IPLS reflected an inhomogeneity in potential origin density, or affinity for firing factors, or both.

Current experimental evidence has not yet clearly distinguished between the origin affinity and origin density models. ORC and MCM abundance profiles, which presumably reflect potential origin density, are well correlated with DNase I HSS and early MRT [9–11]. Furthermore, ORC- or MCM-based IPLSs produced realistic MRT profiles in Gindin-like simulations [10, 11], which supports the origin density model. However, our comparison of ORC, MCM and RFD profiles of the Raji cell line showed that at constant MRT and transcription level, ORC and MCM densities are similar in initiation, random replication and termination zones [9]. Furthermore, a recent landscape of MCM DH footprints revealed that MCM DH

abundance increases from early to late replication domains, which contradicts the origin density model [12]. The two latter studies suggest that potential origins may be more widespread than initiation sites but have different firing efficiencies, perhaps due to specific MCM or histone modifications affecting their affinities for firing factors, in line with the origin affinity model.

In the present work, we harness our previous kinetic model of DNA replication [7] to predict MRT and RFD profiles. Discrete, localized potential origins (MCM DHs), randomly chosen from an arbitrary IPLS before S-phase entry, are activated in a stochastic manner by interaction with limiting firing factors that engage with forks and are released at termination (Fig 1). As each MCM DH is given the same probability to fire, the non-uniformity of the obtained replication profiles only comes from the non-uniformity of the IPLS (origin density model).

Our aim being to extract the IPLS that best predicts available MRT and RFD data, we first compare their information contents. We show a remarkable conformity of MRT and RFD data to a simple mathematical equation that links both profiles. Extending the work by Gindin et al. [42], we then ask whether the correlation of DNase I HSS with origin activation seen at MRT resolution (50–100 kb) still holds true at RFD resolution (< 5 kb). We demonstrate that MRT and RFD data provide distinct information at different scales.

We then train a neural network on simulated MRT and RFD profiles to infer an IPLS that jointly predicts experimental MRT and RFD almost exactly, surpassing IPLSs based on DNase I HSS, ORC, MCM, Bubble-seq, SNS-seq or ORM profiles. In our model, each potential origin has the same intrinsic probability of activation per unit time. The optimised IPLS, which reflects intrinsic origin efficiencies, can be directly compared with ORC and MCM profiles. To compare the IPLS to actual initiation events as monitored by SNS-seq, bubble-seq, OK-seq or ORM, we establish novel mathematical expressions that relate observed and intrinsic origin

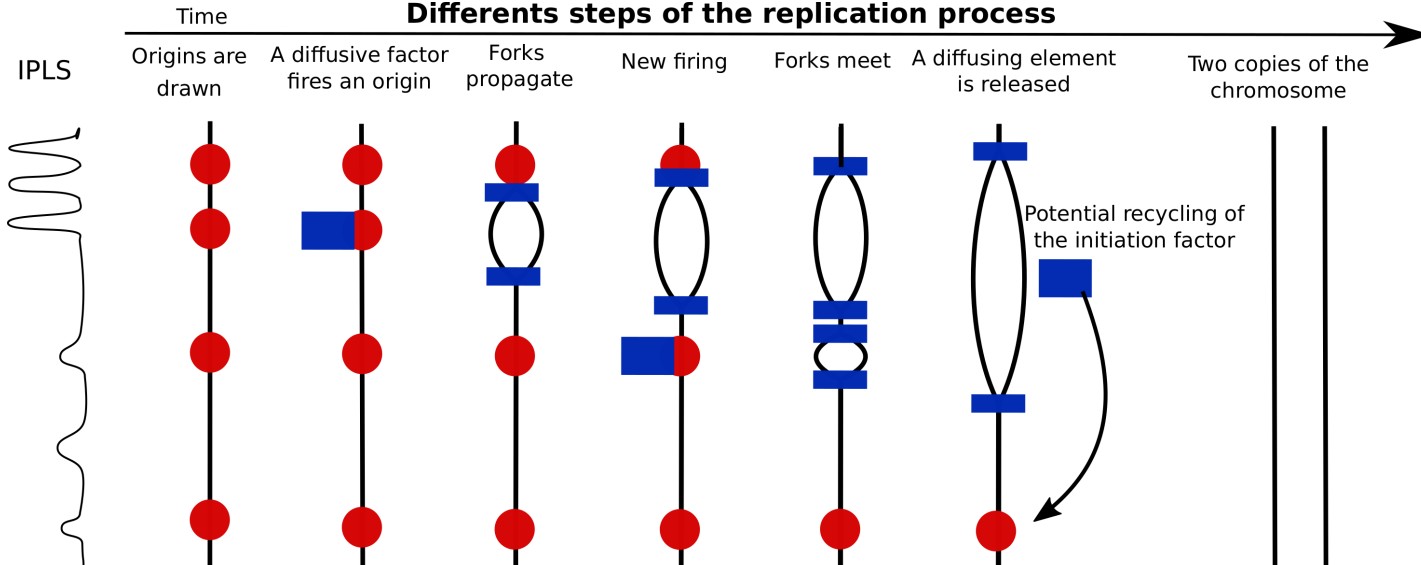

**Fig 1. Modelling DNA replication.** Given an IPLS possibly derived from a specific genomic feature (e.g. a DNase I HSS profile), a fixed number of localized potential origins is drawn (red circles). Limiting firing factors (blue rectangles) activate origins in a probabilistic manner and engage with each pair of newly created forks, which propagate at velocity *v*. Engaged factors can no longer activate origins. Unfired origins are passivated when they are reached by a passing fork. Merging of two forks emanating from adjacent origins results in a replication termination event and the release of one firing factor which becomes available again for origin activation. MRT and RFD are then computed from the average of 200 simulations. See Materials and methods.

efficiencies to MRT and therefore allow us to take origin passivation effects into account. The results show that the optimised IPLS is not fully consistent with ORC abundance profiles, and largely inconsistent with MCM abundance profiles, and that the firing probability of MCM DHs cannot be uniform in time and space. Our results therefore support an origin affinity model and provide a basis to investigate the distinct genetic and epigenetic determinants of origin licensing and firing.

## 2 Results

Note: a nomenclature of all the symbols used in the text is given in the supplementary information (S1 Text).

### Information complementarity between MRT and RFD profiles

Previous modelling works [42, 43] compared simulated and experimental human MRT profiles to constrain their parameter values. RFD profiles have now been established for many human cell lines [21, 22], providing us with an alternative comparison point. It is thus of interest to compare the information content of RFD and MRT profiles. Within the hypothesis of a constant fork speed $v$, MRT and RFD profiles are equivalent as they are analytically related to one another by [44–46]:

$$RFD(x) = v \frac{d}{dx} MRT(x). \tag{1}$$

Here $MRT(x)$ is the mean replication time after entry in S-phase of bin $x$ and is expressed in time units. Note that $MRT(x)$ as measured by Repli-seq experiments is the average global replicated fraction at the moments locus $x$ is replicated, and thus has a value between 0 and 1. To experimentally check Eq (1) we need to convert Repli-seq data, expressed in replicated genome fraction, into time units by multiplication by an estimate of S-phase duration TS. This implicitly assumes a linear relation between replicated fraction and time in S phase, although the true relation appears to be sigmoid rather than linear, which may slightly distort very early or very late S-phase data.

As, the derivative of experimental signal is ill-defined due to noise; slopes cannot be estimated pointwise as they are numerically unstable. This can be circumvented by data smoothing at the expense of resolution by integrating Eq (1) at point $x$ over a length $l$ leading to:

$$\Delta_l MRT(x) = MRT(x + l) - MRT(x) = \frac{1}{v} \int_x^{x+l} RFD(y)\, dy = \frac{l}{v} \langle RFD \rangle_x^{x+l}, \tag{2}$$

where $\langle \cdot \rangle_x^{x+l}$ stands for the average value over $[x, x + l]$. Eq (2) predicts that the MRT change across an interval is proportional to the average RFD over that interval. Using reported Repli-seq MRT [18] and OK-seq RFD [22] profiles for the K562 cell line, Eq (2) was very convincingly verified over scales ranging from 10 kb to 5 Mb, with a genome-wide correlation coefficient up to 0.94 at scale 1.5 Mb, and a proportionality coefficient ranging from $v = 1.2$ kb. min$^{-1}$ to $v = 1.6$ kb.min$^{-1}$, assuming $T_S = 12$ hours [47]. This is illustrated on Fig 2 for scale 50 kb and Fig A in S1 Text for other scales. Therefore, although OK-seq and Repli-seq experiments are complex and have been performed by different laboratories, they are highly consistent with each other, on a wide range of scales, within the hypothesis of a constant fork speed.

In their modelling work, Gindin et al [42] found that of all epigenetic features tested, IPLSs based on DNase I HSS profiles produced the best match between simulated and experimental MRT profiles (Pearson correlation coefficient, PCC = 0.865). We performed similar simulations, using our model (Fig 1) as detailed in Materials and Methods. Using an IPLS based on

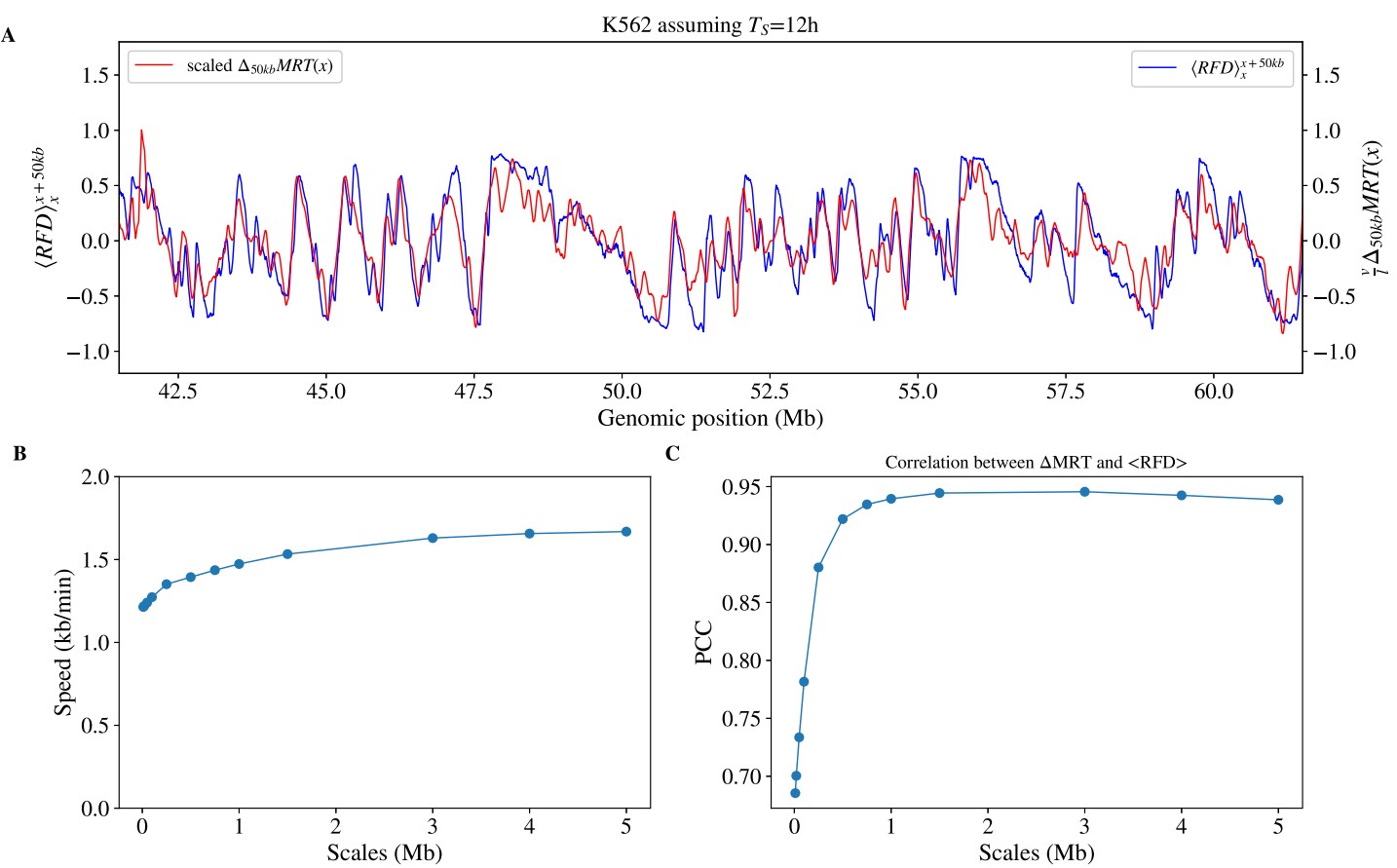

**Fig 2.** (A) Comparison, for a 20 Mb region of chromosome 1, of the K562 RFD profile averaged over 50 kb windows (blue; $\langle RFD \rangle_x^{x+50kb}$) with K562 MRT changes across 50 kb intervals ($\Delta_{50kb}MRT(x)$), following Eq (2) with $v = 1.24$ kb.min$^{-1}$ assuming $T_S = 12$ hour. (B) Replication speed $v$ derived from the proportionality coefficient (Eq (2)) and (C) Pearson correlation coefficient (PCC) between the $\Delta_l MRT(x)$ and $\langle RFD \rangle_x^{x+l}$ profiles at the indicated scales $l$.

the K562 DNase I HSS profile, we drew a fixed number of potential origins and simulated a bimolecular reaction with firing factors, whose number increased from the start of S phase and plateaued after $\approx 1$ h [7, 43]. Productive initiation events trap the factors at the forks and termination events release them, making them available for new initiation events. After grid search optimisation of the number of potential origins and the number of firing factors, we observed a high correlation (PCC = 0.88), similar to Gindin et al ([42]; 0.865), between simulated and experimental MRT profiles, and a lower correlation between simulated and experimental RFD profiles (PCC = 0.70) (Fig 3). Reasoning that addition of dispersed, random initiation to the IPLS (see Methods) might improve the results, we extended the grid search for this parameter and obtained an optimal correlation for RFD at PCC = 0.75 for 5% of random initiation events, while maintaining the same correlation with MRT (PCC = 0.88) (Fig 3). These observations confirm that MRT and RFD data are consistent with each other and suggest that RFD data are furthermore informative about random initiation.

Despite the theoretical equivalence of MRT increments and RFD average values (Eq (2)), their correlation (Fig 2) decreased at small scales, due to the low ($\sim 100$ kb) resolution of MRT profiles. It also decreased, to a lower extent, at large scales, because integrating RFD sums up its experimental noise. In fact, RFD provides better origin position information, while MRT better reflects integrated origin activity over broad regions. This is illustrated by the following numerical experiments.

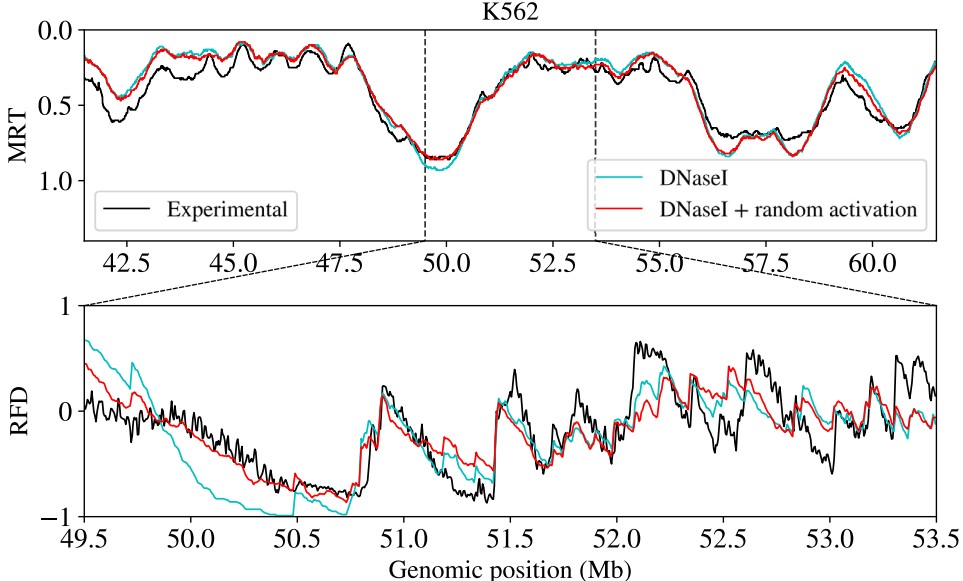

**Fig 3.** Comparison of experimental (black) and simulated (red, light blue) MRT profiles for a ≈ 20 Mb region of chromosome 1 (top) and RFD profiles for a 4 Mb region centered in the middle of the 20 Mb region (bottom) using an IPLS based on DNAse I HSS, with (red) or without (light blue) the addition of 5% of random initiation events. All the parameters of the replication model except the percent of random initiation are the same.

When the positions of DNase I HSS were resampled within each 200 kb window prior to constructing the IPLS, the simulated MRT profile retained a high correlation with the experimental MRT (PCC = 0.87; Fig 4A, green curve), while the correlation between simulated and experimental RFD profiles dropped (PCC = 0.61; Fig 4B, green curve). The exact positions of DNase I HSS were critical to reproduce RFD profiles upward shift positions, in line with the observed enrichment of OK-seq IZs with DNase I HSS [21]. On the other hand, the tolerance of MRT profiles to DNase I HSS resampling suggested that MRT is not sensitive to precise origin positions within a sampling window.

Although MRT can be computed by integrating RFD (Eq (2)), this cumulates the experimental noise, blurring large scale features that MRT data more directly capture. The lesser sensitivity of RFD than MRT to large scale fluctuations in the IPLS was revealed in a second numerical experiment where we modulated the DNase I HSS signal amplitude with a slow varying function of large period ($P$ = 25 Mb) before constructing the IPLS. In that setting, the correlation between simulated and experimental profiles decreased markedly for MRT (PCC = 0.72) but only slightly for RFD (PCC = 0.72) (Fig 4A and 4C, blue curves). Therefore, MRT is constrained by the collective action of multiple origins, so that the sum of neighbouring DNase I HSS signals is critical, while their exact positions per 200 kb windows are not (Fig 4A). RFD is instead sensitive to the location rather than the magnitude of these signals. The influence of a single origin on RFD rapidly decreases with distance, due to the influence of intervening origins.

It may seem illegitimate to make the argument that RFD provides better origin position and MRT better integrated origin activity, on the basis of the magnitude of the drop in PCCs when they do not start at the same level. To address this concern, we performed similar perturbations using DNAse I HSS data as a reference IPLS and the simulated profiles as reference MRT and RFD profiles. When DNase I HSSs were resampled by 200 kb windows, the PCC of unperturbed and perturbed profiles was 0.98 for MRT and 0.67 for RFD. When DNase I HSS

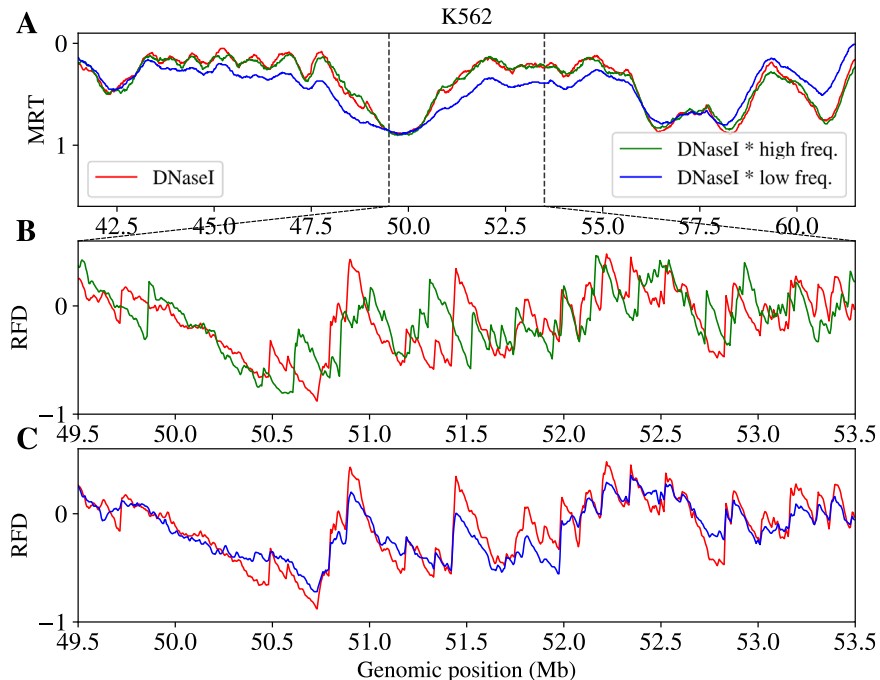

**Fig 4. Comparison of simulated MRT (A) and RFD (B,C) profiles corresponding to different IPLS profiles, all other model parameters being kept constant.** IPLSs were derived from (i) experimental DNase I HSS data (red), (ii) the DNase I HSS data after random shuffling of HS site positions within all 200 kb non-overlapping windows (green), and (iii) DNase I HSS data after modulating their amplitude over a period $P = 25$ Mb (we divided the amplitude signal by $1.1 + \cos(2x/P)$) (blue). DNase I HSS position shuffling in 200 kb windows did not influence simulated MRT profiles but altered RFD profiles significantly, as both red and green signals overlapped in (A) but presented clear differences in (B). Low-frequency modulation of HSS amplitude changed the relative strength of replication timing domains thus altering the MRT profiles, but did not influence the main features of the RFD profile as red and blue signals overlapped in (C) but presented clear differences in (A).

signal was modulated with the slow varying function, the PCC was 0.86 for MRT but 0.93 for RFD. Since in this case the starting PCCs were the same (100%), it can be safely concluded that the first perturbation affects RFD much more than MRT, while the second perturbation affects MRT significantly more than RFD.

To summarize, incorporating RFD as a target for simulations likely allows to test origin positioning at much higher resolution than was achieved with MRT [42, 43]. Deriving RFD profiles from limited resolution ($\sim$ 100 kb) MRT data (Eq (1)) by numerical derivative would produce low resolution RFD profiles, while determining MRT profile from the summation of RFD data (Eq (2)) would produce MRT profiles with unreliable MRT changes over large distances. Experimental MRT and RFD profiles thus provide complementary information. We use both in the following analyses.

**Learning an IPLS that accurately predicts both experimental MRT and RFD data.** Having shown that experimental MRT and RFD profiles are consistent with each other over a wide range of scales at constant fork speed, we assessed to which extent they could be jointly explained by a single IPLS in our replication model. At 5 kb resolution, the IPLS correspond to $\sim$ 575000 parameters which must be optimised. To achieve this, we designed an iterative method that progressively improves the IPLS (Fig 5). It uses model simulations to train a neural network to predict the IPLS given the MRT and RFD profiles, i.e., to invert our replication model. We initialised the method by setting non zero values of the IPLS in regions with the highest RFD derivative (See Materials and methods) i.e. in the strongly ascending segments of

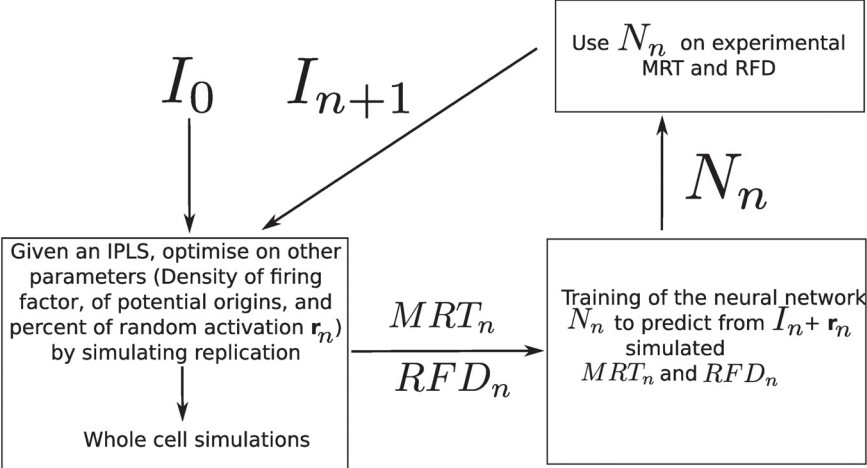

**Fig 5. Scheme of the iterative optimisation procedure of the IPLS for simultaneous prediction of experimental MRT and RFD data.** The starting IPLS $I_0$ may be a crude approximation of the target IPLS such as given by the peaks of RFD derivative, or the DNAse HSS profile, but not a random profile. We observed that the procedure improved the prediction quality after three or four iterations, but not after a larger number of iterations (Table A in S1 Text).

the RFD profile corresponding to the main IZs previously described [21]. This crude approximation of the IPLS is named $I_0$. Then a grid search optimisation on the other parameters $P = (\rho_F, d_{PO}, r)$ of the replication model (See Material and methods) was performed. To limit computation time, this optimisation was performed over chromosome 2 only and resulted in a set of optimal parameters $P_0$ that maximized the sum of the Pearson correlation coefficients between experimental and simulated MRT and RFD profiles. Then we simulated whole genome replication using $I_0$ and $P_0$ to generate $MRT_0$ and $RFD_0$ and trained a neural network (See Materials and methods) to predict $I_0 + r_0$ from $MRT_0$ and $RFD_0$, where $r_0$ is the optimal fraction of random initiation events given $I_0$. We then used this network to predict $I_1$ from experimental MRT and RFD, reasoning that $I_1$ should produce a more accurate prediction of MRT and RFD than $I_0$. Another grid search optimisation on $P$, given $I_1$, was performed to obtain $P_1$ and given $P_1$ and $I_1$ we simulated $MRT_1$ and $RFD_1$. Then a new neural network was trained to predict $I_1 + r_1$ and was then applied to experimental MRT and RFD to obtain $I_2$. These steps were iterated four times, because the correlations between experimental MRT and RFD profiles and their simulated estimates never improved with further iterations (Table A in S1 Text).

We first applied the procedure to K562 data. The sequences of joint correlations between experimental MRT and RFD and simulated profiles ($MRT_0, \cdots, MRT_4$) and ($RFD_0, \cdots, RFD_4$) were (0.81, 0.93, 0.98, 0.98, 0.98) and (0.79, 0.89, 0.91, 0.92, 0.92), respectively. The highest joint correlation was reached at the third iteration and we refer to the maximum initiation potential as $I_M$. We ran a grid search on the whole genome given $I_M$ profile, this yielded unchanged correlation for both MRT and RFD, suggesting that parameter optimisation on chromosome 2 only was not a limitation. We also tried using K562 DNase I HSS as the $I_0$ of the method. The $I_0$ profiles obtained from DNase I HSS or RFD derivative peaks presented some differences (PCC = 0.76), but led to very similar $I_M$ profiles (PCC = 0.94; Fig B in S1 Text) and produced identical high correlations between simulated and experimental MRT (0.98) and RFD (0.91) profiles. In contrast, we were unable to ameliorate the IPLS starting from a random $I_0$ (MRT and RFD correlations were 0.67 and 0.84, respectively, using $I_2$, but decreased at step 3). Therefore, our optimisation method required some initial information

about the IPLS, but converged to nearly the same $I_M$ profile from heterogeneous starting points. This is not a constraint as an adequate initialisation can be obtained from experimental RFD data.

To test the robustness of this inversion procedure, it was applied to replication profiles of GM06990, HeLa and Raji human cell lines and yeast *Saccharomyces cerevisiae*. It systematically resulted in high PCC between experimental and simulated profiles at the third or forth iteration: for MRT 0.99 with GM06990; 0.99 with HeLa, 0.94 with Raji and 0.96 with *S. cerevisiae*; for RFD 0.91 with GM; 0.84 with HeLa; 0.90 with Raji and 0.91 with *S. cerevisiae* (see Table A in S1 Text for the results of the different iterations). Fig 6 illustrates the striking consistency

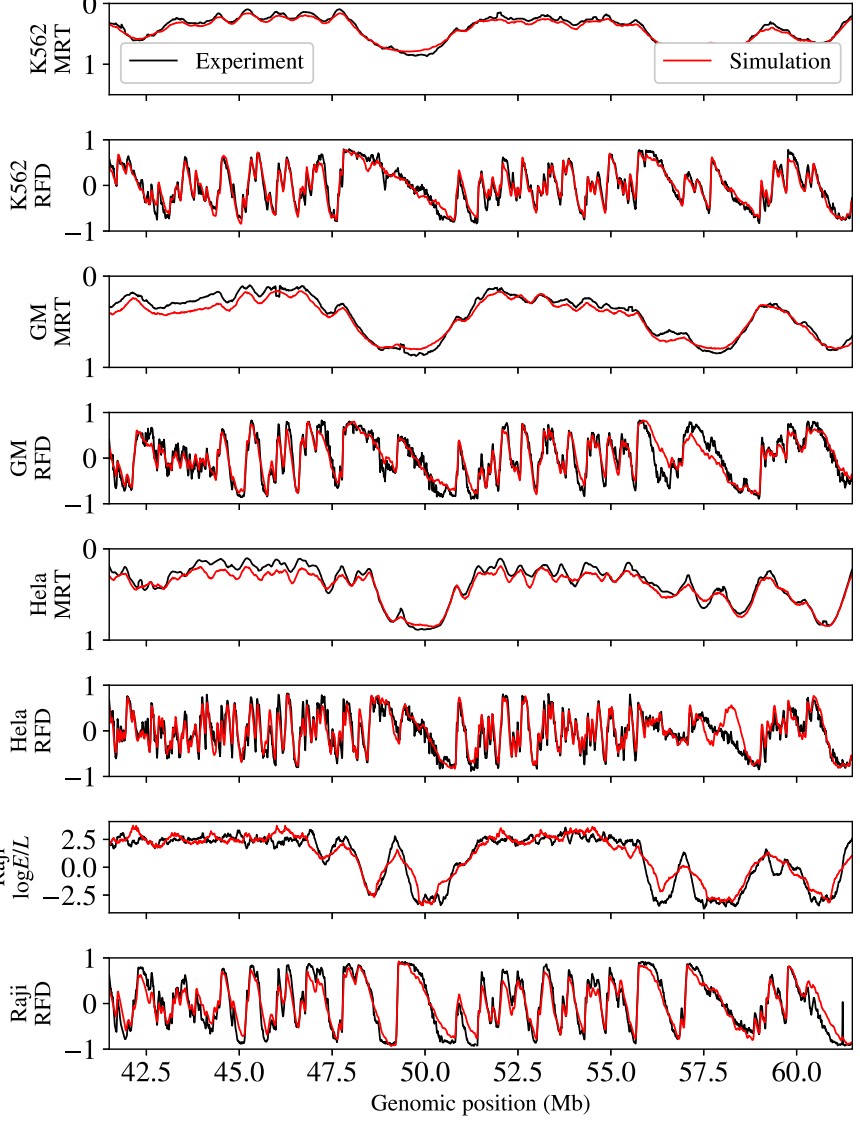

**Fig 6. Comparison for the four indicated cell lines of experimental MRT and RFD (black) and simulated $MRT_M$ and $RFD_M$ profiles (red).** A representative 20 Mb segment of chromosome 1 is shown. Note that the HeLa region of significant discrepancy between experimental and simulated profiles (between 57.7 and 58 Mb) corresponds to a region where experimental MRT and RFD are not coherent: no RFD up-shift matched the MRT peak observed at position 57.6 Mb.

between simulation and experiments obtained for the four different human cell lines. The correlation for Raji MRT was lower than for other cell lines, probably because the Raji Repli-seq profiles were obtained by sorting cells in 2 compartments of S phase instead of 6 for the other cell lines. The correlation for HeLa RFD was lower than for other cell lines. HeLa was more challenging as it has about twice as many IZs as K562, GM06990 and Raji [21, 22]. In addition, regions of poor RFD prediction showed inconsistencies between experimental MRT and RFD, probably due to the use of different Hela cell lines in different laboratories (Fig 6). Fig C in S1 Text shows a zoom on a 10 Mb region of chromosome 1 to illustrate the relationship between MRT, RFD, $I_M$ profile and actual initiation events. The ratio of actual initiation events to $I_M(x)$ increased in late-replicating regions, for reasons elucidated in a later section.

To further quantify the improvement of our predictions over those of Gindin et al [42], we compared their fractions of IPLS and actual initiation events that fell within the IZs reported in Petryk et al [21], which cover 11% and 7% of the HeLa and GM06990 genomes, respectively. Using DNAse I HSS, 35% of the IPLS and 23% of initiation events fell within the HeLa IZs. Using $I_M$ profile, these fractions increased to 46% and 35%, respectively. In GM06990, these fractions increased from 23% to 43% and from 16% to 30%, respectively. Furthermore, we measured the overlap of the strongest 2% upshifts in simulated and experimental RFD profiles. Using $I_M$ profile instead of DNAse I HSS increased the overlap from 11% to 43% in HeLa, from 11% to 51% in GM06990, and from 18% to 60% in K562. These results clearly demonstrate that $I_M$ profile produced markedly better predictions than DNAse I HSS.

**Sensitivity of MRT and RFD with respect to model parameters.** We performed simulations of whole genome replication to assess the sensitivity of MRT and RFD and S-phase duration to: (i) the density of firing factors $\rho_F$ (number per Mb); (ii) the mean distance between potential origins $d_{PO}$; (iii) the proportion of random initiation $r$; and (iv) fork speed $v$. Working around the reference set of parameters (for K562, $\rho_F = \rho_F^* = 0.56$ Mb$^{-1}$; $r = r^* = 0\%$, $d_{PO} = 20$ kb, $v = 1.5$ kb.min$^{-1}$), we let one of the parameter vary (Fig 7). $\rho_F^*$ and $r^*$ are the optimal values obtained at the end of the iterative procedure, without the final grid search exploration on the whole genome as it did not improve the correlations; the values for $d_{PO}$ and $v$ are reasonable choices justified below.

For K562, the correlations of simulated with experimental MRT and RFD profiles showed a clear maximum at $\rho_F = \rho_F^* = 0.56$ Mb$^{-1}$, more pronounced for RFD (Fig 7A and 7B). This number implied that the maximum density of forks was $\sim 1$ fork per Mb, or $\sim 6,000$ forks per diploid nucleus, at any time in S phase. This was in reasonable agreement with upper estimates of the maximal density of limiting factor Cdc45 (3.5 molecules per Mb) in mammalian cells [36], considering that 2 molecules of Cdc45, but only one limiting factor in our model, are required to activate two diverging forks. Similar results were robustly observed using GM06990, Hela and Raji replication data, with maximum PCC values observed for $\rho_F^*$ of 0.51, 0.91 and 0.51 Mb$^{-1}$, respectively (Figs D, E, and F in S1 Text).

Varying $d_{PO}$ and $v$ over a large range of values only weakly changed the correlation of experimental and simulated MRT and RFD (Fig 7D, 7E, 7J and 7K), which justifies why they were left out of the parameter optimisation procedure. We decided to set $v$ to 1.5 kb.min$^{-1}$ and $d_{PO}$ to 20 kb, for the following reasons. First, single molecule studies of DNA replication in human cells have repeatedly reported replication fork speeds of 1–3 kb min$^{-1}$ and distances between activated origins of 50–200 kb [48, 49]. Second, MCM depletion experiments indicated a 5–10 fold excess of loaded MCM DHs over actual initiation events [50]. Third, biochemical quantitation suggested that chromatin is loaded with 1 MCM DH per 20–40 kb at S phase entry [36, 51]. Taken together, these figures are reasonably consistent with each other and with a $d_{PO}$ of 20 kb.

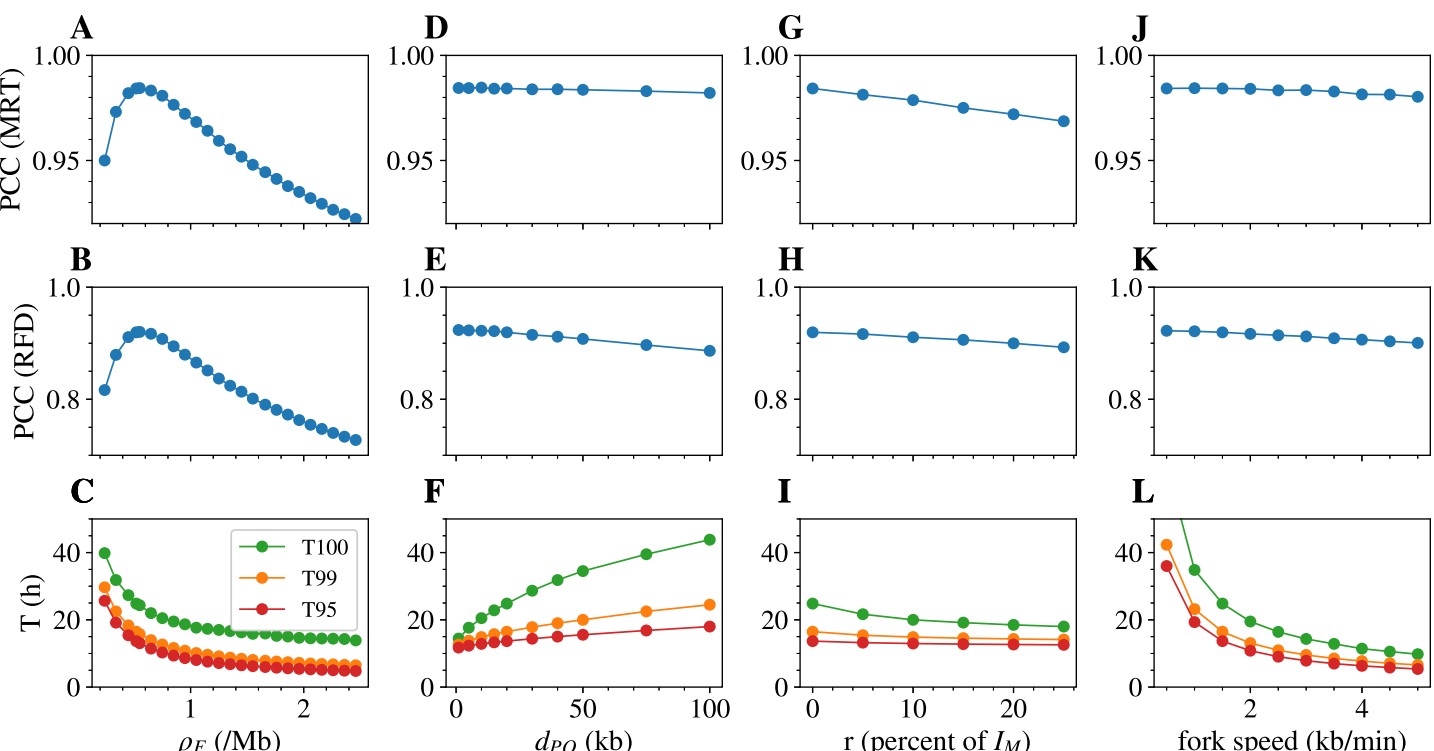

**Fig 7. Effect of single parameter variation on measurable targets in K562, other parameters being kept as their value in the reference parameter set.** Effect of the density of firing factors $\rho_F$ (A,B,C); the average distance between potential origins $d_{PO}$ (D,E,F); the proportion of random initiation $r$ (G,H,I); the fork speed $v$ (J,K,L), on the Pearson Correlation Coefficient (PCC) between simulated and experimental MRT (A,D,G,J) and RFD (B,E,H,K) profiles, and on the median of T95 (red), T99 (orange) and T100 (green), the times required to replicate 95% 99% and 100% of the genome (C,F,I,L).

The optimal value for the random initiation $r^* = 0\%$ was confirmed as increasing $r$ up to 20% slightly decreased both MRT and RFD correlations (Fig 7G–7H), (Figs D, E, and F in S1 Text). The value $r = 0\%$ means that the dispersivity of initiation was correctly learned by the iterative method and did not require further external addition of random initiation. To better apprehend the requirement for inefficient, dispersed initiation, we set to zero the lowest $I_M(x)$ bin values totalling 5% of potential origins ($\approx 53\%$ of the bins in K562). This significantly decreased the correlation with RFD data to PCC = 0.82: without this dispersive, inefficient initiation, the simulations failed to capture the RFD in large late replicating regions whose replication time was also clearly delayed (Fig G in S1 Text). The extended null RFD segments observed in these regions are indeed consistent with random initiation.

To further analyse the distribution of potential and fired origins through the genome, we computed the cumulated share of $I_M$ profile and the cumulated share of actual initiation events, as a function of the genome coverage ranked by increasing $I_M(x)$ values (Fig H in S1 Text). While 80% of $I_M$ profile was concentrated within 15–20% of the genome, this genome fraction only supported 60% of actual initiation events. The remaining 20% of $I_M$ profile gave rise to 40% of actual initiation events that were distributed nearly uniformly over the rest (80–85%) of the genome, with little or no regions completely devoid of initiation events (Fig H in S1 Text). The differential usage of potential origins across the genome arises from the different rate of passive replication, as further elaborated in a later section.

In summary, MRT and RFD data, being highly consistent with each other, were jointly and precisely explained by a simple model featuring a unique $I_M$ profile as input and values for

$d_{PO}$, $v$ and $\rho_F$ in agreement with current knowledge. Dispersive initiation was necessary to fully account for experimental data. Most if not all of the genome showed some basal, non-null replication initiation potential.

## Dependence of replication kinetics on model parameters

We first analyzed K562 S-phase duration $T_S$ using the median of the times to replicate 95%, 99% and 100% of the genome, T95, T99 and T100 respectively. As expected, each of these three times decreased with the density of firing factors $\rho_F$ and the fork speed $v$ (Fig 7C and 7L) as we are in a regime of strong affinity between firing factors and potential origins (large $k_{on}$) so that $T_S \approx \frac{1}{2*v*\rho_F}$ [7]. The model predicted much larger differences between T100 and T99, than between T99 and T95, consistent with the latest replicated regions being the most devoid of potential origins. Indeed, for a genome-averaged distance $d_{PO} = 20$ kb, the predicted distance between potential origins increased from a short 2 kb value in MRT < 0.15 regions to 380 kb in MRT > 0.85 regions (Fig I in S1 Text). This observation also explained that (i) the cell-to-cell variability of T95 or T99 ($\sim$ 10 min) was much smaller than that of T100 (hours) (Fig J in S1 Text); (ii) increasing $d_{PO}$ increased T100 to a much greater extent than T95 or T99 (Fig 7F); and (iii) adding random initiation decreased T100 to a much greater extent than T99 or T95 (Fig 7I). The latter effect decreased with increasing $r$, consistent with the latest replicated regions being fully devoid of origins when $r = 0$ (Fig 7I). Consistently, many late replicating regions showed flat MRT and null RFD profiles characteristic of random initiation [21], but the even later-replicating, common chromosomal fragile sites (CFSs), showed an origin paucity that explains their fragility [52].

Experimentally reported S phase lengths were closer to T95 than T100. Using the reference set of parameters ($\rho_F = \rho_F^*$; $r = r^* = 0\%$, $d_{PO} = 20$ kb, $v = 1.5$ kb.min$^{-1}$), the predicted T95 was 8.6 h for HeLa (experimental estimate 8.8 h; [53]), 13 h in K562 (experimental estimate 12 h; [47]) 13.3 h in Raji, and, assuming a fork speed of $v = 2.0$ kb.min$^{-1}$ as found in the closely related JEFF cell line [49], 10.7 h for GM06990 (experimental estimate 10 h; [44]). One probable explanation is that experimental detection of S phase markers misses the earliest and latest S phase cells, when the replication rate is the lowest. Indeed, very late replication of specific sequences was reported to linger during what is termed the G2 phase of the cell cycle [54], and S phase length variations around the mean ranged from minutes to hours depending on cell lines and detection methods [55, 56]. Within the parameter range we explored, T95 variations ($\sim$ 10 min) were smaller than for T100 (hours) (Fig J in S1 Text). Experiments may therefore have underestimated the exact duration of S phase. Another possibility is that we underestimated $r$ or $v$, since increasing either parameter above its reference value efficiently reduced T100 without much compromising the correlations of simulated and experimental MRT and RFD (Fig 7G, 7H and 7I, Fig D and Fig E in S1 Text). In our simulations, the faster replication of HeLa cells was explained by a larger number of firing factors than in GM06990 (2.0-fold), K562 (1.9-fold) and Raji (1.9-fold). Thus, the optimisation procedure selected a density of firing factors $\rho_F$ that gave relative S phase durations consistent with experimental measurements using sensible values for $d_{PO}$ and $v$.

The origin firing rate per length of unreplicated DNA, $I(t)$, was previously reported to follow a universal bell-shaped curve in eukaryotes [7, 33]. As expected from the choice of the $k_{on}$ value, the simulations produced the expected shape for the four cell lines with a maximum of $I(t)$, $I_{max}$ between 0.01 and 0.02 Mb$^{-1}$.min$^{-1}$, in reasonable agreement with a crude estimate of 0.03–0.3 Mb$^{-1}$.min$^{-1}$ from low-resolution MRT data (Fig K in S1 Text) [33].

Finally, we measured in K562 the dispersion of replication times (RT) of each locus as a function of its MRT. A recent high-resolution Repli-Seq study [19] reported that RT

variability, estimated as the width between the first and the third quartiles of RT distribution, increased from early to mid S phase and decreased thereafter. For most regions the RT variability was in the 1.25–2.5 h range. This study used the replicated genome fraction, as estimated by FACS analysis, as a proxy for RT, and assumed a constant S-phase length of 10 h. When using the replicated genome fraction as a proxy for RT as in [19], our simulations in K562 produced a similar bell-shaped of computed RT variability (Fig L in S1 Text, left). When using the true simulated time, however, the computed RT variability increased throughout S phase, as expected for stochastic origin firing (Fig L in S1 Text, right). These results imply that the bell shape previously reported was an artefact of the non-linear relationship of RT with replicated fraction: the rate of replication is not constant and therefore the replicated fraction is not a suitable proxy for RT to analyse RT dispersion.

In summary, the kinetic parameters of S phase predicted by our stochastic model were (i) consistent with the reported time-dependencies of the firing rate $I(t)$; (ii) consistent with the reported variability of replicated genome fraction at constant MRT; and (iii) predictive of relative S phase durations. The variablity of true RT increased through S phase, as predicted for stochastic origin firing.

## Direct estimation of the IPLS from experimental data

Origin firing can be prevented by context-dependent passivation from nearby origins. It is therefore important to distinguish between the local density of potential origins within a bin, which in the origin density model is directly proportional to the IPLS, and the observed origin efficiency (OE), which is the fraction of chromosomal copies in the cell population in which an origin fired in that bin. OEs can be directly measured by SNS-seq, Bubble-Seq, ORM or indirectly by RFD upshifts. In the following we will derive a mathematical relationship that relates OE to IPLS and MRT. This allows us to predict the IPLS, the input of our simulations, directly from MRT and either SNS-seq, Bubble-Seq, ORM or OK-Seq experiments.

In our model, potential origins are MCM DHs which all have the same elementary probability of firing. For one bin $x$ with $n(x)$ MCM DHs, given the reaction rate $k_{on}$ and the number of free firing factor $F_{free}(t)$, the probability for firing to take place during an elementary time $dt$ is:

$$k_{on}n(x)F_{free}(t)dt \ . \tag{3}$$

Then, the probability $A_x(t)$ for bin $x$ to have been activated at time $t$ without considering passivation is given by:

$$A_x(t) = 1 - e^{-k_{on}n(x)\int_0^t F_{free}(u)du} \ . \tag{4}$$

With an infinite time to fire, $A_x(t)$ converges to one unless the locus is devoid of any potential origins ($n(x) = 0$). Due to the finite length of S phase but also to passivation, the observed efficiency OE is less than one. Replication of a small bin occurs much more often by passivation from nearby origins than by internal initiation (for 5 kb bins, $\Delta RFD/2$, which is an approximation of OE, as discussed later, has a maximum value of 0.17). We can therefore consider that the average passivation time of a bin is not very different from its MRT, here expressed in time that we call $MRT_t(x)$. Thus, $OE(x) = A_x(MRT_t(x))$ which leads to:

$$OE(x) = 1 - e^{-k_{on}n(x)\int_0^{MRT_t(x)} F_{free}(u)du} \ . \tag{5}$$

We first assessed the validity of this relationship in the set of 200 S-phase simulations using $I_M$ profile and the reference set of model parameters in K562. $OE(x)$ and $MRT(x)$ were the

averages of origin firing status and RT recorded in each simulation. The total number of MCM DHs over the genome being $L/d_{PO}$, and the $I_M$ profile being normalised so that its sum is equal to 1, the average MCM DH density profile is $n(x) = I_M(x) * L/d_{PO}$ (MCM DH/5kb).

The two terms of Eq (5) were computed. They showed a genome-wide Pearson correlation coefficient of 0.84 and a proportionality coefficient of 1 (Fig M in S1 Text). When focusing on the local maxima of $\Delta RFD$ ($\sim$ 10, 000 peaks in K562, Materials and Methods), which correspond to IZs, the PCC raised to 0.91, and the proportionality coefficient was 1 (Fig M in S1 Text). These results indicate that our hypothesis that $OE(x) = A_x(MRT_t(x))$ is globally valid but applies even more precisely at IZs determined as $\Delta RFD$ local maxima. However, when exploring different values for $k_{on}$ and $d_{PO}$, we noted that the PCC was stable, but the proportionality coefficient varied from 0.6 to 1.7. This indicates that, for unclear reasons, the validity of Eq (5) is sensitive to the precise combination of origin density and reactivity parameters.

Generally speaking, the $\Delta RFD$ across a genomic segment is twice the difference between the density of initiation and termination events in the segment [46]. Mammalian IZs are broad and may contain multiple MCM DHs. Termination events are nevertheless rare or absent within IZs [21, 28, 57]. Therefore, forks emanating from the first activated MCM DH must rapidly passivate nearby MCM DHs and one can estimate $OE(x) = \Delta RFD(x)/2$ in these loci. For example, the RFD will shift from -1 to +1 across a 100% efficient IZ, creating a jump of $\Delta RFD = 2$. We indeed found a PCC of 0.96 with a proportionality coefficient of 0.92 between $OE(x)$ and $\Delta RFD(x)/2$ at the $\sim$ 10,000 local maxima of $\Delta RFD$ in our K562 simulations (Fig N in S1 Text). As a consequence, Eq (5) linking $OE(x)$, $n(x)$ and $MRT(x)$ provides a link between $RFD(x)$, $n(x)$ and $MRT(x)$ in IZs.

$$n_e(x) = \frac{-\ln(1 - \Delta RFD(x)/2)}{k_{on} \int_0^{MRT_t(x)} F_{free}(u)du} \quad , \tag{6}$$

where $n_e(x)$ is the $n(x)$ profile estimated from the measurable parameters $\Delta RFD$ and MRT. Note that for small value $\Delta RFD(x)$ (for 5 kb bins, the maximum value of $\Delta RFD/2$ is 0.17), one can use a Taylor expansion of ln to simplify Eq (6) yielding:

$$n_e(x) \approx \frac{\Delta RFD(x)}{2k_{on} \int_0^{MRT(x)} F_{free}(u)du} \quad . \tag{7}$$

We then compared $n_e(x)$ estimated using simulated MRT and RFD with $n(x)$, the input of the simulation (Fig 8A). The Pearson correlation was 0.94 and the coefficient of proportionality was 0.70 at the 10,000 $\Delta RFD$ local maxima. Similarly to Eq (5), when exploring different values for $k_{on}$ and $d_{PO}$, the PCC remained stable but the proportionality coefficient varied from 0.4 to 1.2. We also confirmed the $1/\int_0^{MRT(x)} F_{free}(u)du$ dependency (Fig 8B).

The temporal variations of $F_{free}$ are not yet experimentally accessible, but we noticed that $F_{free}$ is approximately constant over a large part of S-phase. (Fig O in S1 Text). Thus, setting $F_{free}(t) = [F_{free}]$ leads to a simple and direct way to estimate the IPLS ($\propto n(x)$) from Repli-Seq MRT and RFD profiles:

$$n_e^a(x) = \frac{-\ln(1 - \Delta RFD(x)/2)}{k_{on}MRT(x)[F_{free}]} \quad , \tag{8}$$

In this case, the latest replicating regions were not well captured by the inverse dependency on $MRT(x)$, but we noticed that the whole range of early and late replicating regions was very

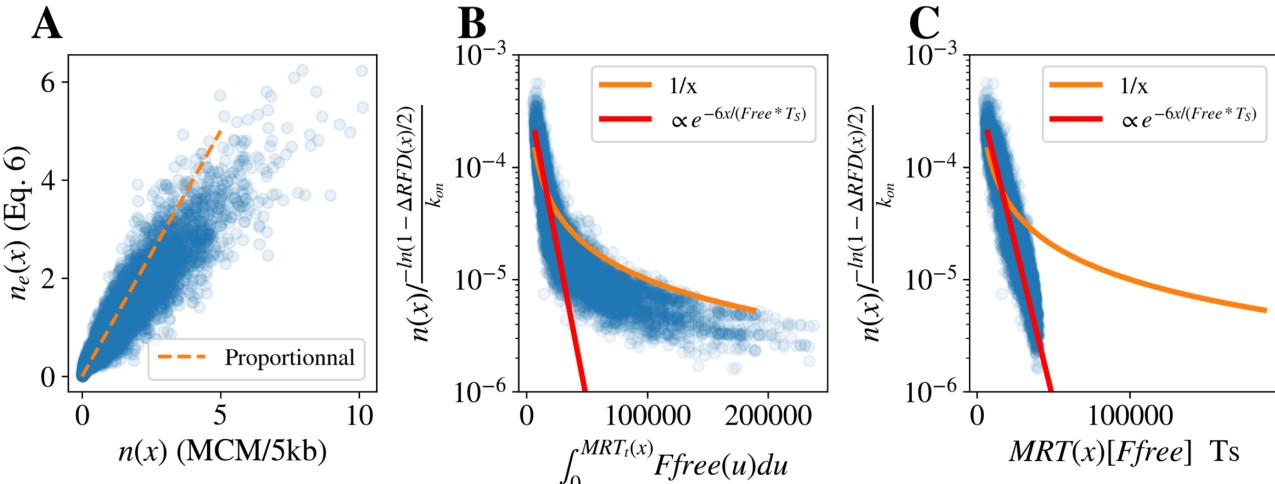

**Fig 8.** (A) Comparison of the average number $n(x)$ of MCM per 5kb bin used in the simulations, and the predicted number of MCM $n_e(x)$ using Eq (6) at the 10,000 $\Delta RFD$ peaks in K562. (B) Scaled $n(x) \times k_{on}/(-\ln(1 - \Delta RFD(x)/2)$ as a function of $\int_0^{MRT_t(x)} F_{free}(u)du$ at the $\sim$ 10,000 $\Delta RFD$ peaks (blue circles) in K562 simulations; it is compared with $1/MRT$ (orange) and proportionality to $e^{-6MRT/T_S}$ (red). (C) Scaled $n(x) \times k_{on}/(-\ln(1 - \Delta RFD(x)/2)$ as a function of Repli-seq $MRT(x)[F_{free}]T_S$ at the 10,000 $\Delta RFD$ peaks (blue circles) in K562 simulations; it is compared with $1/MRT$ (orange) and proportionality to $e^{-6MRT/T_S}$ (red).

well fitted (PCC = 0.97) by an exponential decay (Fig 8C):

$$n_e^{exp}(x) \propto \Delta RFD(x)e^{-6MRT(x)/T_s} \tag{9}$$

We hypothesise that Eq (9) accounts for the actual variations of $F_{free}(t)$ (Fig O in S1 Text) boosting potential origin firing efficiency in late S phase, so that a smaller number of MCM DHs are required to produce the same OEs.

The key advantage derived from Eqs (3–9) is that we are now able using Eqs (8) or (9) to derive an IPLS directly from experimental data alone. Note that since the IPLS is normalized, the ignorance on the prefactors in Eq (9) is not an issue.

For this, we selected the 15% highest experimental $\Delta RFD(x)$ (Materials and Methods) and used Eq (8) to predict an IPLS that we used in our model. After optimisation of the other parameters, the resulting simulated MRT and RFD profiles were highly correlated with experimental profiles (Table 1), for example PCC = 0.94 between MRT and 0.88 between RFD profiles in K562. This formula for IPLS prediction also robustly applied in *S. cerevisae* with PCCs of 0.93 for MRT and 0.90 for RFD. The IPLS derived from Eq (9) led to even higher correlations between simulated and experimental MRT (PCC = 0.97) and RFD (PCC = 0.91), very close to the correlation coefficients obtained using $I_M$ profiles (Table 1). We confirmed this result in *S. cerevisae* with PCC = 0.96 for MRT and PCC = 0.9 for RFD. These results show that combining OEs, estimated by local RFD upshifts, with MRT data, suffices to produce a near optimal IPLS.

In principle, SNS-seq or Bubble-seq signals also provide a direct estimate of OEs, up to a proportionality coefficient. Using such estimates (K562 and HeLa SNS-seq, GM 06998 Bubble-seq) directly as an IPLS resulted in poor correlations between simulated and experimental profiles (MRT, PCC = 0.54, 0.43 and 0.30 and RFD, PCC = 0.16, 0.23 and 0.12, respectively). Combining the same data with MRT data to infer the IPLS using Eq (9) improved the correlations (MRT, 0.81, 0.87 and 0.83 and RFD 0.48, 0.59 and 0.53, respectively) (Table 1), but combining the same MRT information with a flat OE profile produced even better correlations

**Table 1. Best joint correlation between simulated and experimental MRT and RFD data in K562, GM, Hela and Raji cell lines, marginalising over the other parameters of the simulation for different choices of IPLS.** The table was sorted by mean value over the columns.

| Marks | K562 MRT | K562 RFD | GM MRT | GM RFD | Hela MRT | Hela RFD | Raji log $E/L$ | Raji RFD |
|---|---|---|---|---|---|---|---|---|
| $I_M$ | 0.98 | 0.92 | 0.99 | 0.91 | 0.98 | 0.85 | 0.94 | 0.90 |
| $n_e^{exp}$ Eq (9) | 0.97 | 0.91 | 0.97 | 0.88 | 0.96 | 0.83 | – | – |
| $n_e$ Eq (8) | 0.94 | 0.88 | 0.91 | 0.82 | 0.92 | 0.80 | – | – |
| $e^{-6MRT/T_S}$ | 0.97 | 0.71 | – | – | 0.97 | 0.64 | – | – |
| ORM $e^{-6MRT/T_S}$ | – | – | – | – | 0.92 | 0.71 | – | – |
| ORM | – | – | – | – | 0.87 | 0.64 | – | – |
| SNS $e^{-6MRT/T_S}$ | 0.81 | 0.48 | – | – | 0.91 | 0.63 | – | – |
| Bubble $e^{-6MRT/T_S}$ | – | – | 0.83 | 0.53 | – | – | – | – |
| ORC2 | 0.87 | 0.74 | – | – | – | – | 0.46 | 0.16 |
| MCM2 | – | – | – | – | 0.52 | 0.41 | – | – |
| SNS | 0.54 | 0.16 | – | – | 0.43 | 0.23 | – | – |
| ORC3 | – | – | – | – | – | – | 0.50 | 0.16 |
| MCM7 | – | – | – | – | 0.46 | 0.28 | 0.19 | 0.00 |
| Bubble | – | – | 0.30 | 0.12 | – | – | – | – |
| MCM3 | – | – | – | – | – | – | 0.19 | 0.00 |
| hMCM-DH | – | – | – | – | -0.81 | -0.22 | – | – |

(MRT, 0.97, 0.98 and 0.98 and RFD 0.71, 0.62 and 0.71, respectively), suggesting that SNS and Bubble-seq data do not synergize with MRT as favorably as RFD data.

Directly using the HeLa ORM data [24] as an IPLS, we obtained a PCC between predicted and experimental HeLa MRT of 0.87, very close to the PCC reported by Wang et al [24] (0.85). However, the PCC was only 0.64 for RFD. Combining the ORM with MRT data using Eq (9) increased the PCCs for MRT and RFD to 0.92 and 0.71, respectively. Although these numbers were not as good as with $I_M$ profile (0.98 and 0.85, respectively), they were clearly better than with SNS- and Bubble-seq. These results therefore confirm the joint consistency of ORM with Repli-seq and OK-seq data.

We also analysed experimental data on potential origin positioning (Table 1). Using an IPLSs computed from K562 ORC2 [10] resulted in good PCCs between simulated and experimental profiles of 0.87 (MRT) and 0.74 (RFD). Using HeLa MCM2 ChEC-seq [11], MCM7 ChIP-seq [58] or MCM-DH footprints [12], resulted in disparate PCCs of 0.52, 0.46 and -0.81 for MRT and 0.41, 0.28 and -0.22 for RFD, respectively, pointing to strong discrepancies in the three HeLa MCM datasets. Using Raji ORC2 and ORC3 data [9] yielded PCCs of 0.46–0.50 for MRT and 0.16 for RFD, a lesser agreement than for K562 ORC2. Finally, using Raji MCM3 and MCM7 data [9] resulted in PCCs of 0.19 for MRT and 0.00 for RFD. In summary, ORC was a better predictor of the IPLS than MCM. The MRT predicted from MCM DH footprints was strikingly anticorrelated with experimental MRT, due to increasing abundance of MCM DH footprints from early to late replication domains [12]. The predicted RFD profile was also anticorrelated with experimental RFD, but to a lower extent than MRT, probably because short MCM DH clusters aligned well with OK-seq and ORM IZs in early but not late replication domains [12].

Finally, we computed the direct correlation of SNS-seq, Bubble-seq, ORM, ORC and MCM profiles with $I_M$ profile in the cognate cell line at 5 kb and 50 kb resolution (Table 2). The best correlations were obtained for HeLa ORM data (0.51 at 5 kb resolution and 0.62 at 50kb resolution), followed by K562 ORC2 data (0.3 and 0.54, respectively). A significant anticorrelation was observed for HeLa MCM DH footprints (-0.23 and -0.34), consistent with simulation

**Table 2. Correlation between $I_M$ profile and experimental data at 5 kb and 50 kb resolution, as well as RFD upshifts ($I_0$).**

| Marks | K562 (5kb) | K562 (50kb) | GM (5kb) | GM 50kb | Hela (5kb) | Hela (50kb) | Raji (5kb) | Raji (50kb) |
|---|---|---|---|---|---|---|---|---|
| $I_0$ | 0.7 | 0.74 | 0.66 | 0.72 | 0.73 | 0.78 | 0.8 | 0.85 |
| ORM | – | – | – | – | 0.51 | 0.61 | – | – |
| ORC2 | 0.3 | 0.54 | – | – | – | – | 0.2 | 0.24 |
| MCM2 | – | – | – | – | 0.27 | 0.42 | – | – |
| SNS | 0.05 | 0.16 | – | – | 0.18 | 0.26 | – | – |
| ORC3 | – | – | – | – | – | – | 0.21 | 0.24 |
| MCM7 | – | – | – | – | 0.19 | 0.32 | 0.13 | 0.14 |
| Bubble | – | – | 0.06 | 0.15 | – | – | – | – |
| MCM3 | – | – | – | – | – | – | 0.09 | 0.08 |
| hMCM-DH | – | – | – | – | -0.23 | -0.34 | – | – |

results. The correlations of the other datasets to $I_M$ profile ranged from 0.05 to 0.27 and 0.15 to 0.42 at 5 kb and 50 kb resolutions, respectively. Therefore, none of these experimental datasets were convincing predictors of $I_M$, for reasons that remain to be elucidated (Table 2). The fact that ORC better predicted $I_M$ profile than MCM would be unexpected if the multiple MCM DHs loaded by ORC were equally competent to trigger initiation [59, 60]. Assuming this discrepancy does not stem from experimental limitations, it suggests that ORC-proximal MCM DHs are more likely to fire than ORC-distal ones.

In our model, all origins have the same $k_{on}$, and the spatial dependency is encoded in the non uniform density of potential origins. Since the effective reactivity of a potential origin is proportional to $k_{on}F_{free}$, the observed differences between experimental MCM density and $I_M(x)$ may be explained by spatial or temporal non-uniformity, i.e. locus-dependent $k_{on}$ or time-dependent $F_{free}$, with the $k_{on}F_{free}$ landscape given by the $I_M(x)/MCM(x)$ ratio. We found that the normalized $I_M(x)/MCM(x)$ ratio computed in 50 kb windows in HeLa cells (excluding null MCM windows; Fig P in S1 Text) decreased with MRT but was broadly dispersed even at constant MRT. A similar trend was observed with Raji MCM data even though the decrease was less progressive. This global trend could be explained if firing factor abundance decreased during S phase, as recently reported [61], but the broad dispersion also implied that even at similar MRT, all MCM DHs were not equally reactive to firing factors, possibly due to MCM DH modifications or to chromatin environment. Finally, we cannot exclude that experimental noise or differential MCM loading dynamics during G1 [62] prevent an accurate picture of MCM distribution at S phase entry, especially considering the significant discrepancies observed between the three HeLa MCM datasets. Interestingly, the K562 $I_M(x)/ORC(x)$ ratio did not decrease with MRT but was still broadly dispersed at constant MRT Fig P in S1 Text). These results suggests that although more MCM DHs are loaded per ORC in late than in early replicating regions, this does not result in increased MCM firing in late regions, possibly due to above-discussed mechanisms and an increased firing propensity of ORC-proximal MCM DHs. Although the Raji data showed a much more similar behavior of $I_M(x)/ORC(x)$ and $I_M(x)/MCM(x)$ ratios, quantitations reported in [9] were also consistent with a slight increase of MCM/ORC ratio in late compared to early replicating regions.

According to Eq (6), the faster passivation of early than late IZs means that early IZs would require several-fold more MCM DHs than late IZs to achieve a similar OE. In our simulation with optimised $k_{on}$, $F_{free}$ and $d_{PO}$, the maximum RFD upshift per 5kb bin was 0.17, an OE that would require as much as $\sim$ 20 MCM DHs per 5 kb if an early MRT of 1h is to be achieved. Given that MCM DHs occupy $\sim$ 60 bp each and are only found in internucleosome linkers

[11], a 5 kb chromatin segment may not accomodate more than 25 MCM DHs. This steric limit is almost reached (see also Fig I in S1 Text), suggesting that additional regulatory mechanisms that increase $k_{on}$ or $F_{free}$ in early S phase may be required to boost the intrinsic firing efficiency of some MCM DHs, consistent with Fig P in S1 Text. The higher abundance of MCM DH in late than in early-replicating domains reported by [12] advocates for mechanisms that not only boost the firing efficiency of early MCM DHs clusters aligned with early OK-seq and ORM IZs, but also delay the firing of the more expanded and denser MCM DH clusters observed in later replicating domains.

To summarize, we checked that OEs can be accurately measured from RFD upshifts and we could predict from these OEs and MRT the number of potential origins (MCM DHs) per 5 kb bin in IZs assuming the specific value for $d_{PO}$ used in our model, where all MCM DHs have the same locus- and time-independent probability of firing per unit time. However, the observed discrepancies between predicted and observed MCM DH densities, and the steric MCM loading constraint discussed above, support a mixed, potential origin density and affinity model where MCM DHs may have different affinities for firing factors. Future investigations of the $I_M(x)/MCM(x)$ and $I_M(x)/ORC(x)$ ratios should help reveal the licensing and post-licensing mechanisms that regulate origin firing probability.

## Discussion

Understanding the spatiotemporal programme of DNA replication in human cells has been hampered by inconsistencies in origin mapping studies and the complexities of their genetic and epigenetic determinants. Here, we demonstrate a high reciprocal consistency of MRT and RFD profiles and provide novel methods to infer an optimal IPLS, $I_M$, that jointly predicts MRT and RFD with nearly complete accuracy in multiple cell lines. $I_M$ profile represents a significant improvement over the previous IPLS based on DNAse I HSSs. Importantly, $I_M$ profile predicts that a significant amount of dispersive initiation is required to optimally account for MRT and RFD. In our modelling framework, the IPLS is equivalent to a potential origin density landscape where all potential origins have the same elementary probability of firing, independently of genomic location and time. The best molecular candidates for potential origins are the MCM DH complexes and/or their chromatin loader ORC. However, ORC binding profiles were good but not perfect predictors of $I_M$ profile, and MCM DH profiles were poorly or even negatively correlated to $I_M$. In a distinct approach, we devised novel methods to estimate the IPLS from the distribution of actual replication initiation events mapped by SNS-seq, Bubble-seq or ORM. ORM, but not SNS-seq and Bubble-seq, led to good predictions of MRT and RFD profiles. The reciprocal consistency of MRT, RFD and ORM is remarkable given current limitations of ORM resolution (15 kb). The inability of all MCM genomic profiles to correctly predict MRT and RFD implies that the assumption that all MCM DHs have the same elementary probability of firing (strict origin density model) must be abandoned in favor of a variable origin affinity model. However, the high variability of MCM profiles prevents us to totally exclude that when the source of this variability is understood, an improved MCM profile may better predict RFD and MRT.

Eq (1) implies that, without noise, MRT and RFD profiles in a constant fork speed hypothesis account for the same information. Eq (2) shows that one can compare MRT increments with the integral of RFD using only one free parameter, the fork speed. The narrow range of fitted fork speed values (from $\approx 1.2$ to $\approx 1.6$ kb/min) over the large range of scales explored (5 kb to 5 Mb), and the high correlations obtained (from 0.68 to 0.95), show that MRT and RFD are mutually consistent, even at lower resolutions than expected (MRT resolution $\approx 100$kb) and that the hypothesis of a constant fork speed at resolutions down to 5 kb is robust. Our

numerical experiments involving resampling of DNaseI HSSs furthermore show that RFD contain higher resolution information than MRT, whereas MRT contain information about integrated initiation strengths of large domains, that is lost when integrating RFD due to noise. We therefore used both profiles to invert these data into IPLSs.

Mathematical models have been previously developed to estimate intrinsic origin efficiencies from MRT [63, 64], and RFD [65]. The models assign either a discrete number of origins [63, 65], each having a time-dependent probability of firing, or a continuous spatiotemporal initiation density [64]. These hypotheses are more complex than ours and make it difficult to discriminate between origin density and affinity models. Furthermore, these methods either require a non trivial optimisation that may be feasible with the yeast but not the human genome, given its size, or a Bayesian analysis that so far was limited to sets of 3 origins [65]. In contrast, our neural network approach is very flexible, can easily accommodate new datasets and is fast even with the human genome. Furthermore, it outputs a 1D profile of potential origin density in human cells, which can be directly compared to experimental origin licensing profiles, as first achieved in yeast [40].

Further examination of $I_M$ profile show that although 80% of the IPLS is concentrated within 15–20% of the genome, this genome fraction only gave rise to 60% of initiation events, the remaining 40% being nearly uniformly distributed over the rest of the genome. These dispersive initiation events were essential to fully account for MRT and RFD data. Their prevalence was not much different from that inferred in Miotto [10] (60%) or measured at two model chicken loci by FISH-combing [28]. Genome-wide single molecule analyses reported a lower but significant 10–20% of dispersed initiation events in yeast [26, 27].

Reasoning that an origin's passivation typically occurs at its MRT, we derived a novel mathematical relationship (Eq (5)) that, assuming a constant availability of firing factors through S phase, relates the numbers of potential origins to observed efficiencies (OEs) and MRT. OEs can be estimated by RFD upshifts and, the IPLS being proportional to the number of potential origins, Eq (8)) allows in turn to derive an IPLS. The predicted IPLS is proportional to the RFD upshift and to $1/MRT$ (Eq (8)). This was fairly well verified in simulated datasets, but we empirically found that an $1/e^{6MRT}$ dependency (Eq (9)) gave even better predictions, probably because the recycling of firing factors by termination events increases in late S phase. Strikingly, the IPLS inferred from experimental RFD and MRT data using Eq (9) generated almost as good simulated profiles as the $I_M$ inferred using neural networks. We however caution that this procedure is sensitive to smoothing and thresholding and therefore less robust that neural networks.

Eqs (5) to (9) imply that, in the origin density model, early IZs require several-fold more MCM DHs than late IZs to achieve a similar OE. This may potentially explain why SNS-seq and Bubble-seq were poorly correlated to $I_M$ profile. We therefore used Eq (9) to exploit the MRT information and convert these OE measurements into IPLS. The resulting IPLS was improved yet still poorer than inferred from a flat OE profile. We conclude that SNS-seq and Bubble-seq data are not consistent with MRT and RFD data. When the same approach was used with ORM data, however, a remarkable consistency with MRT and RFD data was observed.

Assuming that ORC and MCM profiles are not all biased, our results impose to relax the assumption that all MCM DHs are equally reactive to firing factors, or the assumption that the concentration of firing factors is constant. Examination of the $I_M(x)/ORC(x)$ and $I_M/MCM(x)$ ratios suggests that, while more MCMs per ORC are loaded in late- than in early-replicating DNA, MCM DH activation probability is higher in early-replicating DNA and next to ORC, and varies along the genome even at constant MRT. Mechanisms that locally increase MCM firing propensity may allow early IZs to reach a high IPLS without increasing MCM DH

density beyond steric constraints. Mechanisms that locally decrease MCM firing propensity in the dense, expanded MCM DH clusters observed in late replicating domains [12] may keep them inactive until late S phase yet capable of fast replication when inhibition is lifted.

Studies in yeast reported several mechanisms that regulate origin usage after MCM DH loading. The Ctf19 kinetochore complex [66] and the forkhead transcription factors Fkh1/2 [67] both facilitate origin firing by local recruitment of Cdc7-Dbf4, a kinase required for MCM DH phosphorylation mandatory for origin firing. Conversely, Rif1, a global regulator of replication timing, locally recruits the PP1 phosphatase to counteract Cdc7-Dbf4-mediated phosphorylation of MCM DH [68–70]. Furthermore, the Rpd3 histone deacetylase delays the firing of many origins by modulation of Fkh1-origin binding and, hence, DDK recruitment to origins [71].

Recent studies also imply Rif1-PP1 in replication timing regulation in higher eukaryotes [72, 73]. In Drosophila embryos, Rif1 hubs form at satellite DNA sequences to prevent early origin firing and undergo switch-like dispersal in response to a changing balance of Rif1-PP1 and Cdc7-Dbf4 activities, resulting in a very late but synchronous replication of satellite DNA [73], reminiscent of the late and random replication mode of human heterochromatic segments [21] loaded with dense, expanded MCM DH clusters [12].

A study of cell-cycle regulated MCM turnover suggested that parental MCMs (inherited from the mother cell after mitosis) are favored over nascent (newly synthesized) MCMs to trigger initiation events, possibly due to distinct post-translational modifications associated with chromatin engagement during the previous cell cycle [74]. Furthermore, it was recently reported that origin dormancy and activation are regulated by distinct post-translational MCM modifications that reflect a balance between the activity of the SIRT1 deacetylase and checkpoint signaling by the ATR kinase [75].

The chromatin context may also regulate origin firing. IZs are enriched in DNAse I hypersensitive sites (HSSs) and histone modifications typical of active transcriptional regulatory elements [21]. Whereas chromatin openness may conceivably facilitate origin licensing, the study of Li et al [12] suggests that chromatin accessibility is not a limiting factor and is even anticorrelated to MCM DH loading. We therefore envision that chromatin openness may rather facilitate the origin firing step, consistent with the association of open chromatin marks with binding sites for the limiting firing factor MTBP [31]. Understanding the genetic and epigenetic determinants of MCM DH density and reactivity to firing factors remains an exciting goal for future studies.

## Materials and methods

### Model and simulation

**Model and parameters.** We model the replication initiation process by a bimolecular reaction between free firing factors $F_{Free}$ and potential origins $PO$ on the genome of size $L$ with a reaction rate $k_{on}$. This rate is the probability per unit of time that a firing factor and a $PO$ meet and that an initiation follows. Once an initiation event has occurred, two forks propagate in opposite direction at speed $v$ and a firing factor is trapped. The number $N_F$ of firing factor is fixed and parametrised as $N_F = \rho_F L$ with $\rho_F$ the density of firing factors. A potential origin density landscape $I$ is used to position $\frac{L}{d_{PO}}$ origins prior to each S phase entry along the genome, with $d_{PO}$ the genome average distance between potential origins. We also decompose the initiation probability landscape $I$ in two terms: it is the sum of an inhomegeneous profile $I_e$, for example derived from an epigenetic landscape, and a uniform contribution which correspond to a proportion $r$ of the initial profile $I_e$. In all the simulations we fixed $v = 1.5$ kb/min (unless otherwise stated) and $k_{on} = 3\,10^{-6}$ min$^{-1}$. A typical simulation therefore has 4 free parameters $(\rho_F, \rho_{PO}, I_e, r)$.

**Simulation implementation.** The modeled genome is always considered at 5 kb resolution. The input spatial profile $I$ is normalised so that the sum is one. We draw from this normalised profile $N_{PO}(t=0) = \frac{L}{d_{PO}}$ origins, with replacement, meaning that several origins can be drawn in the same 5 *kb* window. During the simulation we introduce $\rho_F L$ firing factors following an exponential characteristic law $\rho_F L(1 - e^{-t/\tau})$ with $\tau$ a characteristic time taken to 1 h to simulate progressive activation of firing factor upon S-phase entry. We use a Gillespie algorithm [76] to simulate an initiation reaction with reaction rate $k_{on}N_{PO}(t)N_F(t)$ with $N_F(t)$ the number of free firing factor at time $t$. The Gillespie algorithm considers that the the next reaction between an origin and a firing factor will take place after a time $\delta t_A$ drawn from an exponential distribution of parameter $\frac{1}{k_{on}N_{PO}(t)N_F(t)}$. Then $\delta t_A$ is compared to $\delta t_E$, the smallest time of encounter for two forks on the genome. The system then evolve for an increment of time $\delta t = min(\delta t_E, \delta t_A)$, meaning that all the forks on the genome moves of $v\delta t$. If $\delta t_A < \delta t_E$ then one origin is activated at random, a firing factor is trapped ($N_F(t + \delta t) = N_F(t) - 1$), and two forks propagate on opposite directions at a velocity $v$ from the origin. Otherwise the termination event releases a factor so that $N_F(t + \delta t) = N_F(t) + 1$ and a new time step begin. If a fork replicates a position with unfired origins, then these origins are passivated and remove from the list of potential origins.

**Remark on rescaling the simulation for the simulation on chromosome 2 only.** In [7], we showed that to reproduce the shape of the experimental temporal rate of DNA replication origin firing, one has to be in a regime governed by the critical parameter $\rho_F^* = \frac{v}{k_{on}Ld_{PO}}$. In order to stay in this regime no matter the size of the genome, we in fact parameterized the simulations with $k_{on}^e = k_{on}L$ chosen constant. $k_{on}^e = 8.625$ kb min$^{-1}$, so that when we simulated the whole genome (in our case, the first 22 human chromosomes whose total size is 2 875 Mb), then $k_{on} = 310^{-6}$ min$^{-1}$. This means that if we decrease the size of the system, the constant of reaction $k_{on}$ increases, which is coherent as $k_{on}$ encompass the efficiency of encounter between one potential origin and one firing factor and being in a smaller system their encounter rate is increased.

**Missing data.** For all simulations if a gap larger than 1.5 Mb without data in either MRT or RFD experimental data was present, the region extended of 500 kb on both ends was removed. This mainly happen in telomeric and centromeric regions, and it means that a chromosome can be segmented in two or more pieces. Then when comparing e.g. simulated with experimental MRT, we also remove all gaps in MRT data that are smaller than 1.5 Mb extended of 500 kb on both ends. These two steps remove less than 10% of the genome for either MRT or RFD in all considered cell lines. When computing replication time, we exclude all gaps present in either MRT or RFD data as well their surrounding 500 kb.

**Computing experimental quantities, comparison with experimental data.** To compute RFD, we record for each 5 kb window in each simulation the fork direction as +1, −1 or 0 if an initiation or a termination occurred. Then $RFD = (n_R - n_L)/200$, where $n_R$ (resp. $n_L$) are the number of times the fork direction was +1 (resp. −1) in the 200 simulations. To compute the MRT as done in Repli-seq experiments, we recorded for each simulation and each locus the actual replicated fraction of the genome at the time the locus is replicated. Then to simulate the six fractions of the Repli-seq experiment, the continuous [0..1] interval of replicated fraction of the genome is mapped to six bin of length 1/6. Then $MRT(x) = \sum p_i(x)i/6 + 1/12$ where $p_i(x)$ if the fraction of the simulations where the locus at position $x$ has been replicated when the replicated fraction of the genome was between [$i/6, (i + 1)/6$] ($i \in \{0, \cdots, 5\}$).

For both MRT and RFD when comparing with experimental data, we masked the region removed as specified in the missing data paragraph. Pearson correlations were computed at 5 kb resolution for RFD and 10 kb resolution for MRT. $T_{100}$ was defined as the replication time of the latest replicated window. T99 (resp. T95) is defined as the time at which 99% (resp. 95%) of the genome was replicated.

## Experimental data

DNaseI HS data were downloaded from the ENCODE project [18, 77]: (K562 DNaseI HS Uniform Peaks from ENCODE/Analysis (table wgEncodeAwgDnaseUwdukeK562UniPk.narrowPeak).

ORC2 binding sites in K562 cells were obtained from [10] (supplementary material table S1 in [10]).

SNS-seq data were obtained for K562 from [13] (GSE46189_Ori-Peak.bed) and for Hela from [78] (Hela_SNS_seq_Besnard_Tot.1kb.csv).

Bubble-Seq data were obtained from [14] (GSE38809_GM_combined_RD_bubbles.bedgraph).

RFD profiles derived from OK-seq data were obtained for Hela from [21] and for GM06990, K562 and Raji from [22]. After mapping the fragments, the RFD is computed as follow: $RFD = (R - F)/(R + F)$ where R (resp. F) is the number of reads mapped to the reverse (resp. forward) strand of the considered regions.

For mean replication timing data, GM12878, K562, HeLaS3, alignment files of Repli-seq libraries (BAM files) for six S-phase fractions were obtained from the ENCODE project [18, 77] at http://hgdownload.cse.ucsc.edu/goldenPath/hg19/encodeDCC/wgEncodeUwRepliSeq/.

For the MRT profile in Raji, we used the early- to late-RT ratio determined by Repli-seq [79] as pre-computed in supplementary file GSE102522_Raji_log2_hg19.txt downloaded from GEO (accession number GSE102522).

For yeast, RFD profiles derived from OK-seq data were obtained from [27] and MRT was obtained from [80]. We shifted MRT by 0.05 for numerical stability.

hMCM-DH data [12] was aligned on hg19. Due to the presence of peaks of extremely high intensity the signal was thresholded at the 99th percentile.

ORC2, ORC3, MCM2 and MCM7 profiles in Raji cell line were obtained from [9].

MCM2 Hela data [11] was aligned on hg19. Due to the presence of peaks of extremely high intensity the signal was thresholded at the 99th percentile.

ORM data [24] were obtained directly from the authors.

## Grid search optimisation

When performing grid search optimisation on the four parameters $\rho_F$, $d_{PO}$, $r$, $v$ we noticed, as explained in the main text, that both $d_{PO}$ and $v$ had little effect on the MRT and RFD profiles. These two parameters where thus left out so that $\rho_F$ and $r$ optimisation was carried on a 2-dimensional grid. The optimum selected is the one of highest sum of Pearson correlation between simulated and experiment MRT and RFD. For the grid search optimisation carried on chromosome 2 (second Results' section), the explored $r$ values were [0,0.02,0.05,0.1], and $\rho_F$ values were [0.6, 0.7, 0.8, 0.9, 1., 1.1, 1.2, 1.3, 1.4]$\times\rho_{start}$ where $\rho_{start}$ was defined as the firing factor density needed to replicate the whole genome at a fork speed of 1.5 kb.min$^{-1}$ in an S-phase duration $T_S$, if all the firing factors are active: $\rho_{start} = \frac{1}{2*v*T_S}$, with $T_S$ = 8 h for Hela cell and 12 h for GM06990, K562 and Raji.

When performed genome-wide (Results reported in Table 1), we chose the same grid for all human cell lines. $r$ varied from 0 to 20% by increments of 5%, and explored $\rho_F$ values were [0.27, 0.41, 0.55, 0.68, 0.82, 0.95, 1.1] Mb$^{-1}$.

## Iterative procedure used in learning the IPLS that best predicts both MRT and RFD data

To define a starting initiation profile, we computed the RFD increments between 5 kb bins, smoothed the profile using a 50 kb average sliding window, kept values higher than the 80th

percentile and set the rest to zero. Using this profile we ran a grid search optimisation (see previous paragraph) to obtain $MRT_0$ and $RFD_0$ that best fitted experimental data. Then a neural network $N_1$ was trained to predict the initiation profile $I_0 + r$ ($r$ being the amount of random activation obtained from the grid search optimisation part) from the simulated $MRT_0$ and $RFD_0$ (see next paragraph for details on the neural network). $N_1$ was then applied on experimental $MRT$ and $RFD$ predicting the $I_1$ profile, which was then used in a simulation to obtain $MRT_1$ and $RFD_1$ and compute the correlation with experimental $MRT$ and $RFD$. We reiterated the process twice using successively $MRT_1$ and $RFD_1$ as input and then the obtained $MRT_2$ and $RFD_2$ as input to produce $MRT_3$ and $RFD_3$. We reiterated the procedure once more and stopped as it did not improve the correlations. The code to reproduce these steps is available at (https://github.com/organic-chemistry/repli1D).

### Neural network training

The input of the network was a window of size 2005 kb (401 bins of 5 kb) with both MRT and RFD, an the output was the initiation signal at the center of the window. The RFD was smoothed with a 50 kb rolling window. We used a three-layer convolutional neural network with kernel length 10 and filter size 15 and a relu activation. Each layer was followed by a dropout layer at a value of 1%. The last convolutional layer was followed by a maxpooling of kernel size 2. The resulting vector went through a dense layer with sigmoid activation and output size 1, meaning that the 2005 kb window allowed to compute the probability of activation at its center. We used a binary cross entropy loss and the layer was trained with the adadelta algorithm. The procedure was implemented using keras. We used chromosome 3 to 23 for the training, chromosome 2 for validation and chromosome 1 for testing. To make the network more robust to experimental noise, we randomly added noise to the input RFD profile by assigning 1% of the bins a random value between -1 and 1.

### Analytical extraction of $n_e$ from $\Delta RFD$ and $MRT$ data

Eqs (7)–(9) requires that $\Delta RFD$ be estimated. We smoothed human $RFD$ data using a running average window of 15 kb (3 bins) then computed $\Delta RFD$ between consecutive 5kb windows. The 15% top values were selected and the other bins set to 0. The non-zero bins were divided by $1/\int_0^{MRT(x)} F_{free}(u)du$, $1/MRT$ or by $e^{-6MRT}$.

### Selecting $\Delta RFD$ peaks

We ran a peak detection algorithm over $\Delta RFD$ at a 5 kb resolution, using scipy routine find_-peaks with parameters width = 4 and height = 0.02, thus selecting peaks $\geq$4x5kb in width and $\geq 0.02$ $\Delta RFD$/5kb in height. This yielded 9878 peaks for GM06990, 13466 peaks for Hela and 10009 peaks for K562. The selected peaks as well as $\Delta RFD$ and $I_M$ profile for comparison are shown on Fig Q in S1 Text.

### Smoothing of a profile

When we refer to smoothing at a given $x$ scale in the text, we refer to a running average on $x$ kb window centered on the middle of the window. At chromosome ends or internal gaps in the data the window is truncated to the available data.

### Supporting information

**S1 Text. Table A.** Pearson correlation coefficients between experimental MRT and RFD profiles and their simulated $MRT_n$ and $RFD_n$ estimates obtained for the series of iteratively

optimised IPLS $I_n$ using RFD derivative for initialisation of $I_0$. Results are shown for K562, GM06990, Hela and Raji cell lines as well as *S. cerevisiae*. At the 5$^{th}$ iteration none of the PCC increased (not shown). **Fig A.** Comparison between $\langle RFD \rangle_x^{x+l}$ (blue) and $\frac{v}{l}\Delta_l MRT(x)$ (red) (Eq (2)) assuming $T_S = 12h$ at different scale $l$. From (top) to (bottom), $l = 100$ kb, 200 kb, 500 kb and 1000 kb and fork speed values $v$ are taken from Fig 2B. The same 20 Mb region of chromosome 1 as in Fig 2A is shown. **Fig B.** Comparison of K562 IPLSs optimised either from DNase I HS peaks (orange) or from the experimental $\Delta RFD$ peaks (blue). The obtained $I_M$ profiles are almost identical (PCC = 0.94). **Fig C.** Zoom on chromosome 1 for K562. Top: experimental (black) and simulated (red) MRT. Middle Top, experimental black) and simulated (red) RFD. Middle Bottom, input $I_M$ profiles. Bottom, Probability of activation. **Fig D.** Effect of single parameter variation on measurable targets in GM06990. Effect of the density of firing factors $\rho_F$ (A,B,C); the average distance between potential origins $d_{PO}$ (D,E,F); the percent of random initiation $r$ (G,H,I); the fork speed $v$ (J,K,L), on the Pearson Correlation Coefficient (PCC) between simulated and experimental MRT (A,D,G,J) and RFD (B,E,H,K) profiles, and on T95 (red), T99 (orange) and T100 (green), the median times required to replicate 99% and 100% of the genome (C,F,I,L). **Fig E.** Effect of single parameter variation on measurable targets in Hela. Effect of the density of firing factors $\rho_F$ (A,B,C); the average distance between potential origins $d_{PO}$ (D,E,F); the percent of random initiation $r$ (G,H,I); the fork speed $v$ (J,K,L), on the Pearson Correlation Coefficient (PCC) between simulated and experimental MRT (A,D,G,J) and RFD (B,E,H,K) profiles, and on n T95 (red), T99 (orange) and T100 (green), the median times required to replicate 99% and 100% of the genome (C,F,I,L). **Fig F.** Effect of single parameter variation on measurable targets in Raji. Effect of the density of firing factors $\rho_F$ (A,B,C); the average distance between potential origins $d_{PO}$ (D,E,F); the percent of random initiation $r$ (G, H,I); the fork speed $v$ (J,K,L), on the Pearson Correlation Coefficient (PCC) between simulated and experimental log2 E/L (A,D,G,J) and RFD (B,E,H,K) profiles, and on n T95 (red), T99 (orange) and T100 (green), the median times required to replicate 99% and 100% of the genome (C,F,I,L). **Fig G.** (Top) Comparison of experimental MRT (black) with simulated MRT using the optimised IPLS $I_M$ (red) and after setting to zero the bins with the lowest $I_M(x)$ values corresponding to 5% of the total origin firing events ($\approx 53\%$ of the bins) (blue). (Bottom) Same comparison for the RFD profiles. **Fig H.** Left: fraction of the total IPLS as a function of the coverage for genome sorted from low to high initiation. Right: fraction of the total probability of activation as a function of the coverage for genome sorted from low to high IPLS. **Fig I.** Predicted number of potential origins per kb, computed for each 5 kb bin, vs. MRT in K562 replication simulations. **Fig J.** Distribution of T95 (blue),T99 (orange) and T100 (green) replication times as defined in the text for the four different cell lines, as indicated. **Fig K.** Probability of initiation per length of unreplicated DNA per minute for the four indicated cell lines. **Fig L.** Replication time (RT) variability as a function of MRT in simulated K562 replication. Blue dots and orange bars indicate the genome-wide average and range of values, respectively of RT variability. A) RT variability computed using the replicated genome fraction as a proxy for RT, plotted against MRT. B) RT variability using the true simulated time, plotted as a function of MRT. **Fig M.** Comparison of observed origin efficiency in K562 replication simulation, directly counted as the fraction of simulations in which replication started in a bin (OE), or computed as the right-hand side term of Eq (5), genome wide (left) or restricted to the peaks of $\Delta RFD$ (right). Red line represents the linear fit and the orange line the first diagonal. **Fig N.** Comparison of OE, directly counted as the fraction of simulations in which replication started in a bin (OE), with its estimation by $\Delta RFD/2$ restricted to peaks of $\Delta RFD$. The orange line is the first diagonal. **Fig O.** Number of Free firing factors as a function of the S-phase fraction for the four indicated cell lines. **Fig P.** A) $I_M(x)/ORC_2(x)$ ratio in K562. B) $I_M(x)/MCM_2(x)$ ratio in

Hela. C)$I_M(x)/MCM_7(x)$ ratio in Hela. D)$I_M(x)/hMCM\text{-}DH(x)$ ratio in Hela. E) $I_M(x)/ORC_2(x)$ ratio in Raji. F) $I_M(x)/ORC_3(x)$ ratio in Raji G) $I_M/MCM_3$ ratio in Raji H) $I_M(x)/MCM_7(x)$ ratio in Raji. Due to noise in MCM or ORC data all signals were smoothed with a 50 kb sliding window and normalized so that the median value over all the genome was one (orange line). For A-D, the black dotted lines indicate the median value of the ratios by 0.05 MRT steps. For E-H, the black dotted lines indicate the median value of the ratios by 1 log $E/L$ steps. **Fig Q.** Peak detection of $\Delta RFD$ (dots on the top of the orange signal) overlayed with $I_M$ profile (blue curves) for the three indicated cell lines.
(PDF)

## Acknowledgments

We thank CL Chen and D Saulebekova for kindly providing the ORM data. We thank the Centre Blaise Pascal, O. Gandrillon and the IN2P3 computing center for providing the computing resources.

## Author Contributions

**Conceptualization:** Jean-Michel Arbona, Olivier Hyrien, Benjamin Audit.

**Data curation:** Hadi Kabalane, Jeremy Barbier.

**Funding acquisition:** Olivier Hyrien.

**Methodology:** Jean-Michel Arbona, Arach Goldar, Olivier Hyrien, Benjamin Audit.

**Project administration:** Jean-Michel Arbona.

**Software:** Jean-Michel Arbona.

**Supervision:** Benjamin Audit.

**Writing – original draft:** Jean-Michel Arbona, Olivier Hyrien, Benjamin Audit.

**Writing – review & editing:** Jean-Michel Arbona, Hadi Kabalane, Jeremy Barbier, Arach Goldar, Olivier Hyrien, Benjamin Audit.

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
