## [Editor Report · Decision Letter 0]

23 Nov 2022

Dear Dr Arbona,

Thank you very much for submitting your manuscript "Neural network and kinetic modelling of human genome replication reveal replication origin locations and strengths" for consideration at PLOS Computational Biology.

As with all papers reviewed by the journal, your manuscript was reviewed by members of the editorial board. We also transferred the reviewes from your previous submission to eLife. In light of these reviews, we would like to invite the resubmission of a significantly-revised version that takes into account the reviewers' comments. Please, refer to the original reviews you received from eLife.

We cannot make any decision about publication until we have seen the revised manuscript and your response to the reviewers' comments. Your revised manuscript is also likely to be sent to reviewers for further evaluation.

Sincerely,

Andrea Ciliberto

Academic Editor

PLOS Computational Biology

Ilya Ioshikhes

Section Editor

PLOS Computational Biology
---

## [Decision Letter · Decision Letter 1]

31 Mar 2023

Dear Dr Arbona,

Thank you very much for submitting your manuscript "Neural network and kinetic modelling of human genome replication reveal replication origin locations and strengths" for consideration at PLOS Computational Biology. As with all papers reviewed by the journal, your manuscript was reviewed by members of the editorial board and by several independent reviewers. The reviewers appreciated the attention to an important topic. Based on the reviews, we are likely to accept this manuscript for publication, providing that you modify the manuscript according to the review recommendations.

Sincerely,

Andrea Ciliberto

Academic Editor

PLOS Computational Biology

Ilya Ioshikhes

Section Editor

PLOS Computational Biology

Reviewer's Responses to Questions

**Comments to the Authors:**

Reviewer #1: The authors have thoroughly and satisfactorily addressed the concerns raised in my previous review.

Reviewer #2: For the previous version of this paper found the reviewers appreciated the usefulness of the approach and the quality of the results but also raised collectively a number of concerns, ranging from technical comments on methods to the reliance on relatively low-quality MCM mapping data in drawing conclusions about the replication program. In the revised version, the authors have done a very good job of responding substantively to the points raised and have improved significantly the paper (likely increasing its impact). The methods of inferring IPLS given here are interesting and fruitful, even if all aspects (such as the form of Eq. (9)) are not fully understood. I suspect that the conclusions as to the variability of MCM firing propensity will continue to be debated, but this paper will be the starting point for any future discussion. I thus recommend publication in PLOS Computational Biology after the following minor points are addressed:

Line 173: “Assuming a linear relation between replication fraction and S-phase duration…”

Why? All of the types of models considered have a sigmoidal relation between the S-phase time and replication fraction. Indeed, the discussion in Lines 436-450 concerns just this point….

Line 178: “… circumvented by data smoothing…”

This is precisely what Eq. (2) does (at a length scale l)

Line 251: “deriving RFD profile from MRT data Eq. (1) by numerical derivative would produce low resolution RFD profiles with amplified noise….”

— again, not true if the derivatives were estimated in a more sophisticated way.

For Lines 178, 251, there is nothing wrong with methods used and I agree that MRT is better at long and RFD at short scales, but it’s also true that there is no real advantage of using Eq. (2) rather than Eq. (1) with derivatives smooth over the same length scale.

Line 505: Taylor expansion (not extension)

Lines 517-520 and Eq. (9): The authors’ arguments are superficially reasonable, and I am willing to let the point stand. However, the good agreement between the ad hoc exponential in Eq. (9) seen in Fig. 8C, supposedly from combining the time-variation of Ffree(t) with the expected 1/t dependence, makes one wonder….

• It might be clearer to write I_M(x) rather than I_M to emphasize that the profile is something that varies along the genome, as opposed to other quantities such as fork velocity that are assumed constant over the genome. (The x-dependence is occasionally given, as in Line 481, but not usually.)

• The phrase “reciprocally consistent” should probably be “mutually consistent” as the former suggests MRT and RFD might be inversely related, which is not what is intended.

Reviewer #3: In the revised manuscript the authors have addressed most of the concerns that were raised in the initial review. The overall performance of their models is impressive, and their results provide a solid foundation on which other researchers can build.

However, I suggest that the authors:

1) discuss the inconsistencies of the measurements of ORC and MCM components and the difficulties that these inconsistencies create when attempting to draw conclusions about the relative contributions of origin licensing vs firing and

2) clarify the role of “confounding parameters” such as transcription and MRT when discounting origin density model.

1. One of the goals of the study is to help elucidate the relative contribution of origin density vs. origin affinity in shaping genome replication in human cells. While the authors' replication modeling performs well at both large (MRT) and smaller scales (RFD), the final determination of the origin affinity vs origin density critically depends on having accurate measurements of ORC and MCM density across the genome. In Table 1, correlations for ORC are as high as 0.87 (for MRT) and 0.74 (for RFD) in K562 cells and lower but still positive at 0.46 and 0.16 in Raji cells. For members of the MCM complex such correlations were all over the place: from relatively high positive correlations: 0.52 (MRT) and 0.41 (RFD) for MCM2 in HeLa cells; no correlations 0.19 (MRT) and 0.00 (RFD) for MCM3 in Raji; to high negative correlations -0.80 (MRT) and -0.22 (MRT) for hMCM-DH. Which one of these MCM measurements is true? The authors recognize robust correlations of ORC density with both MRT and RFD based on high positive correlations for ORC2 in HeLa cells but dismiss the correlation between MCM and MRT and RDF even though the correlations for MCM2 are higher at 0.52 (MRT) and 0.41(RFD) in HeLa cells compared to ORC2 correlations of 0.46(MRT) and 0.16(RFD) in Raji cells. Given inconsistencies in MCM measurements, it is difficult to draw firm conclusions about licensing vs firing models based on these results and the authors should discuss these limitations.

2. Lines 123 to 125

The authors write: “However, our comparison of ORC, MCM and RFD profiles of the Raji cell line showed that when confounding parameters such as MRT and transcription status are controlled, ORC and MCM densities are not predictive of IZs.”

Based on the absence of correlations of MCM3 and RFD in Raji cells R=0.00, it is not surprising that MCM3 does not delineate IZs in their previous study (Kirstein et al 2021), whether or not confounding parameters are taken into account. On the other hand, replication correlated with MCM density based on MCM2 density measurements (Foss 2021) for both MRT (0.52) and RFD(0.41), which could perhaps explain higher MCM density in IZs in that study.

Finally, why would one need to take transcription status into account in determining origin density vs origin affinity model. If MCM densities were the sole determinant of replication initiation, and the MCM densities are a reflection of transcription (i.e. MCM are not found within transcribed genes), removing transcription as a “confounding parameter” would immediately discount the origin density model.

**Have the authors made all data and (if applicable) computational code underlying the findings in their manuscript fully available?**

Reviewer #1: Yes

Reviewer #2: Yes

Reviewer #3: Yes

PLOS authors have the option to publish the peer review history of their article (what does this mean?). If published, this will include your full peer review and any attached files.

Reviewer #1: No

Reviewer #2: No

Reviewer #3: No

Figure Files:

Data Requirements:

Reproducibility:

References:

---

## [Decision Letter · Decision Letter 2]

30 Apr 2023

Dear Dr Arbona,

We are pleased to inform you that your manuscript 'Neural network and kinetic modelling of human genome replication reveal replication origin locations and strengths' has been provisionally accepted for publication in PLOS Computational Biology.

Best regards,

Andrea Ciliberto

Academic Editor

PLOS Computational Biology

Ilya Ioshikhes

Section Editor

PLOS Computational Biology

Reviewer's Responses to Questions

**Comments to the Authors:**

Reviewer #2: The authors have satisfactorily addressed my remaining concerns, and I recommend publication in its present form for this very nice contribution.

Reviewer #3: The authors have addressed all my concerns.

**Have the authors made all data and (if applicable) computational code underlying the findings in their manuscript fully available?**

Reviewer #2: Yes

Reviewer #3: Yes

PLOS authors have the option to publish the peer review history of their article (what does this mean?). If published, this will include your full peer review and any attached files.

Reviewer #2: No

Reviewer #3: No

---

## [Editor Report · Acceptance letter]

25 May 2023

PCOMPBIOL-D-22-01492R2 

Neural network and kinetic modelling of human genome replication reveal replication origin locations and strengths

Dear Dr Arbona,

I am pleased to inform you that your manuscript has been formally accepted for publication in PLOS Computational Biology. Your manuscript is now with our production department and you will be notified of the publication date in due course.

With kind regards,

Anita Estes
